# PROGRESS MEASURES FOR GROKKING VIA MECHANISTIC INTERPRETABILITY

**Neel Nanda**[*,†]     **Lawrence Chan**[‡]     **Tom Lieberum**[†]     **Jess Smith**[†]     **Jacob Steinhardt**[‡]

## ABSTRACT

Neural networks often exhibit emergent behavior, where qualitatively new capabilities arise from scaling up the amount of parameters, training data, or training steps. One approach to understanding emergence is to find continuous *progress measures* that underlie the seemingly discontinuous qualitative changes. We argue that progress measures can be found via mechanistic interpretability: reverse-engineering learned behaviors into their individual components. As a case study, we investigate the recently-discovered phenomenon of "grokking" exhibited by small transformers trained on modular addition tasks. We fully reverse engineer the algorithm learned by these networks, which uses discrete Fourier transforms and trigonometric identities to convert addition to rotation about a circle. We confirm the algorithm by analyzing the activations and weights and by performing ablations in Fourier space. Based on this understanding, we define progress measures that allow us to study the dynamics of training and split training into three continuous phases: memorization, circuit formation, and cleanup. Our results show that grokking, rather than being a sudden shift, arises from the gradual amplification of structured mechanisms encoded in the weights, followed by the later removal of memorizing components.

## 1 INTRODUCTION

Neural networks often exhibit emergent behavior, in which qualitatively new capabilities arise from scaling up the model size, training data, or number of training steps (Steinhardt, 2022; Wei et al., 2022a). This has led to a number of breakthroughs, via capabilities such as in-context learning (Radford et al., 2019; Brown et al., 2020) and chain-of-thought prompting (Wei et al., 2022b). However, it also poses risks: Pan et al. (2022) show that scaling up the parameter count of models by as little as 30% can lead to emergent reward hacking.

Emergence is most surprising when it is abrupt, as in the case of reward hacking, chain-of-thought reasoning, or other phase transitions (Ganguli et al., 2022; Wei et al., 2022a). We could better understand and predict these phase transitions by finding *hidden progress measures* (Barak et al., 2022): metrics that precede and are causally linked to the phase transition, and which vary more smoothly. For example, Wei et al. (2022a) show that while large language models show abrupt jumps in their performance on many benchmarks, their cross-entropy loss decreases smoothly with model scale. However, cross-entropy does not explain *why* the phase changes happen.

In this work, we introduce a different approach to uncovering hidden progress measures: via *mechanistic explanations*.[1] A mechanistic explanation aims to reverse engineer the mechanisms of the network, generally by identifying the circuits (Cammarata et al., 2020; Elhage et al., 2021) within a model that implement a behavior. Using such explanations, we study *grokking*, where models abruptly transition to a generalizing solution after a large number of training steps, despite initially overfitting (Power et al., 2022). Specifically, we study modular addition, where a model takes inputs $a, b \in \{0, \dots, P-1\}$ for some prime $P$ and predicts their sum $c \bmod P$. Small transformers trained with weight decay on this task consistently exhibit grokking (Figure 2, Appendix C.2).

---

[*]Corresponding author, please direct correspondence to: `neelnanda27@gmail.com`

[†]Independent researcher.

[‡]University of California, Berkeley.

[1]Interactive versions of figures, as well as the code to reproduce our results, are available at `bit.ly/progress-measures-grokking-website`.

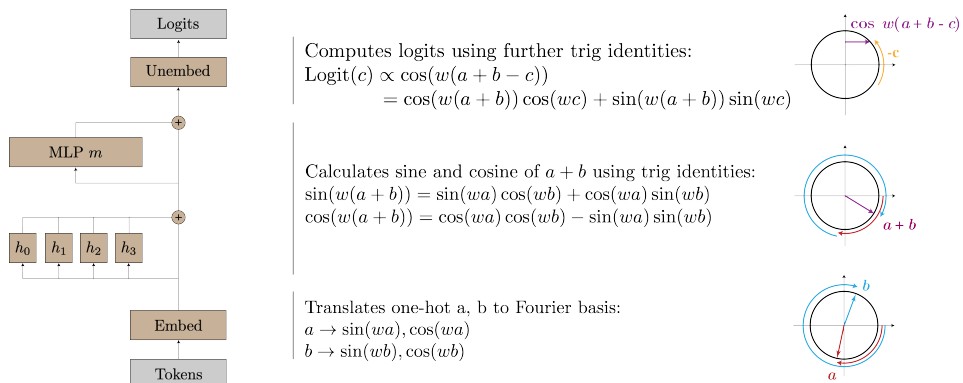

Figure 1: The algorithm implemented by the one-layer transformer for modular addition. Given two numbers $a$ and $b$, the model projects each point onto a corresponding rotation using its embedding matrix. Using its attention and MLP layers, it then composes the rotations to get a representation of $a + b \mod P$. Finally, it "reads off" the logits for each $c \in \{0, 1, ..., P - 1\}$, by rotating by $-c$ to get $\cos(w(a + b - c))$, which is maximized when $a + b \equiv c \mod P$ (since $w$ is a multiple of $\frac{2\pi}{P}$).

We reverse engineer the weights of these transformers and find that they perform this task by mapping the inputs onto a circle and performing addition on the circle. Specifically, we show that the embedding matrix maps the inputs $a, b$ to sines and cosines at a sparse set of *key frequencies* $w_k$. The attention and MLP layers then combine these using trigonometric identities to compute the sine and cosine of $w_k(a + b)$, and the output matrices shift and combine these frequencies.

We confirm this understanding with four lines of evidence (Section 4): (1) the network weights and activations exhibit a consistent periodic structure; (2) the neuron-logit map $W_L$ is well approximated by a sum of sinusoidal functions of the key frequencies, and projecting the MLP activations onto these sinusoidal functions lets us "read off" trigonometric identities from the neurons; (3) the attention heads and MLP neuron are well approximated by degree-2 polynomials of trigonometric functions of a single frequency; and (4) ablating key frequencies used by the model reduces performance to chance, while ablating the other 95% of frequencies slightly *improves* performance.

Using our understanding of the learned algorithm, we construct two progress measures for the modular addition task—*restricted loss*, where we ablate every non-key frequency, and *excluded loss*, where we instead ablate all key frequencies. Both metrics improve continuously prior to when grokking occurs. We use these metrics to understand the training dynamics underlying grokking and find that training can be split into three phases: **memorization** of the training data; **circuit formation**, where the network learns a mechanism that generalizes; and **cleanup**, where weight decay removes the memorization components. Surprisingly, the sudden transition to perfect test accuracy in grokking occurs during cleanup, *after* the generalizing mechanism is learned. These results show that grokking, rather than being a sudden shift, arises from the gradual amplification of structured mechanisms encoded in the weights, followed by the later removal of memorizing components.

## 2 RELATED WORK

**Phase Changes.** Recent papers have observed that neural networks quickly develop novel qualitative behaviors as they are scaled up or trained longer (Ganguli et al., 2022; Wei et al., 2022a). McGrath et al. (2021) find that AlphaZero quickly learns many human chess concepts between 10k and 30k training steps and reinvents human opening theory between 25k and 60k training steps.

**Grokking.** Grokking was first reported in Power et al. (2022), which trained two-layer transformers on several algorithmic tasks and found that test accuracy often increased sharply long after achieving perfect train accuracy. Millidge (2022) suggests that this may be due to SGD being a random walk on the optimal manifold. Our results echo Barak et al. (2022) in showing that the network instead makes continuous progress toward the generalizing algorithm. Liu et al. (2022) construct small examples of grokking, which they use to compute phase diagrams with four separate "phases" of learning. Thilak et al. (2022) argue that grokking can arise without explicit regularization, from an optimization anomaly they dub the slingshot mechanism, which may act as an implicit regularizer.

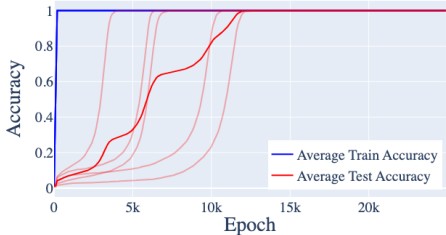 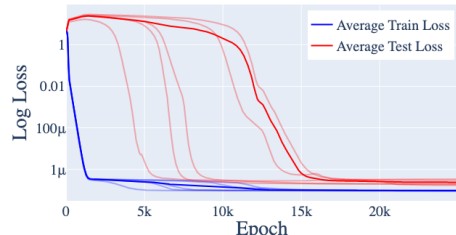

Figure 2: The train and test accuracy (left) and train and test loss (right) of one-layer transformers on the modular addition task described in Section 3, over 5 random seeds. These models consistently exhibit grokking: they quickly overfit early on in training, but then later learn to generalize.

**Circuits-style mechanistic interpretability.** The style of post-hoc mechanistic interpretability in Section 4 is heavily inspired by the Circuits approach of Cammarata et al. (2020), Elhage et al. (2021), and Olsson et al. (2022).

**Progress measures.** Barak et al. (2022) introduce the notion of *progress measures*—metrics that improve smoothly and that precede emergent behavior. They prove theoretically that training would amplify a certain mechanism and heuristically define a progress measure. In contrast, we use mechanistic intepretability to discover progress measures empirically.

## 3 SETUP AND BACKGROUND

We train transformers to perform addition mod $P$. The input to the model is of the form "$a$ $b$ =", where $a$ and $b$ are encoded as $P$-dimensional one-hot vectors, and = is a special token above which we read the output $c$. In our mainline experiment, we take $P = 113$ and use a one-layer ReLU transformer, token embeddings with $d = 128$, learned positional embeddings, 4 attention heads of dimension $d/4 = 32$, and $n = 512$ hidden units in the MLP. In other experiments, we vary the depth and dimension of the model. We did not use LayerNorm or tie our embed/unembed matrices.

Our mainline dataset consists of 30% of the entire set of possible inputs (that is, 30% of the $113 \cdot 113$ pairs of numbers mod $P$). We use full batch gradient descent using the AdamW optimizer (Loshchilov & Hutter, 2017) with learning rate $\gamma = 0.001$ and weight decay parameter $\lambda = 1$. We perform $40,000$ epochs of training. As there are only $113 \cdot 113$ possible pairs, we evaluate test loss and accuracy on all pairs of inputs not used for training.

Networks trained on this task consistently exhibit grokking. As Figure 2 shows, our networks first overfit the training set: train accuracy quickly converges to 100% and the train loss quickly declines, while the test accuracy remains low and the test loss remains high. After around $10,000$ epochs, the network generalizes and test accuracy increases to near 100%. In robustness experiments, we confirm that grokking consistently occurs for other architectures and prime moduli (Appendix C.2). In Section 5.3 we find that grokking does not occur without regularization.

To describe transformer components, we follow the conventions and notations laid out in Elhage et al. (2021). We focus on the $d \times p$ embedding matrix $W_E$, the $d \times n$ output matrix of the MLP layer $W_{out}$, and the $P \times d$ unembedding matrix $W_U$.[2] Let Logits$(a, b)$ denote the logit vector on inputs $a, b$, and $MLP(a, b)$ denote the MLP activations. Empirically, our networks do not significantly use the skip connection around the MLP (Appendix A.1), so Logits$(a, b) \approx W_U W_{out}$MLP$(a, b)$. We therefore also study the $P \times n$ neuron-logit map $W_L = W_U W_{out}$.

### 3.1 THE FOURIER MULTIPLICATION ALGORITHM

We claim that the learned networks use the following algorithm (Figure 1):

- Given two one-hot encoded tokens $a, b$ map these to $\sin(w_k a)$, $\cos(w_k a)$, $\sin(w_k b)$, and $\cos(w_k b)$ using the embedding matrix, for various frequencies $w_k = \frac{2k\pi}{P}, k \in \mathbb{N}$.

---

[2]We ignore the embedding and unembedding of the '=' token for simplicity.

- Compute $\cos(w_k(a+b))$ and $\sin(w_k(a+b))$ using the trigonometric identities:
$$\cos(w_k(a+b)) = \cos(w_k a)\cos(w_k a) - \sin(w_k a)\sin(w_k b)$$
$$\sin(w_k(a+b)) = \sin(w_k a)\cos(w_k b) + \cos(w_k a)\sin(w_k b)$$
In our networks, this is computed in the attention and MLP layers.
- For each output logit $c$, compute $\cos(w_k(a+b-c))$ using the trigonometric identity:
$$\cos(w_k(a+b-c)) = \cos(w_k(a+b))\cos(w_k c) + \sin(w_k(a+b))\sin(w_k c). \quad (1)$$
This is a linear function of the already-computed values $\cos(w_k(a+b))$, $\sin(w_k(a+b))$ and is implemented in the product of the output and unembedding matrices $W_L$.
- The unembedding matrix also adds together $\cos(w_k(a+b-c))$ for the various $k$s. This causes the cosine waves to constructively interfere at $c^* = a+b \mod p$ (giving $c^*$ a large logit), and destructively interfere everywhere else (thus giving small logits to other $c$s).

We refer to this algorithm as Fourier multiplication, and will justify our claim in detail in Section 4.

## 4    REVERSE ENGINEERING A ONE-LAYER TRANSFORMER

In this section, we describe four lines of evidence that our transformers are using the Fourier multiplication algorithm described in Section 3.1. Here we apply our analysis to the mainline model from Section 3; the results are broadly consistent for other models, including across different number of layers, different fractions of the training data, and different prime moduli (see Appendix C.2, especially Table 5).

Our first line of evidence involves examining the network weights and activations and observing consistent **periodic structure** that is unlikely to occur by chance (Section 4.1). Moreover, when we take Fourier transforms, many components are either sparse or nearly sparse in the Fourier domain, supported on a handful of *key frequencies*.

We next look into the actual **mechanisms** implemented in the model weights (Section 4.2). We show that the unembedding matrix $W_L$ is (approximately) rank 10, where each direction corresponds to the cosine or sine of one of 5 key frequencies. Projecting the MLP activations onto the components of $W_L$ approximately produces multiples of the functions $\cos(w_k(a+b))$ and $\sin(w_k(a+b))$, showing that the MLP layer does compute these sums.

To better understand the mechanism, we **zoom in** to individual neurons (Section 4.3). We find that the attention heads and most neurons are well-approximated by degree-2 polynomials of sines and cosines at a *single* frequency. Moreover, the corresponding direction in $W_L$ also contains only that frequency. This suggests that the model's computations are (1) localized across frequencies and (2) mostly aligned with the neuron basis.

Finally, we use **ablations** to confirm that our interpretation is faithful (Section 4.4). We replace various components of the model by the components of the Fourier multiplication algorithm and find that doing so consistently does not harm and sometimes even improves model performance.

### 4.1    SUGGESTIVE EVIDENCE: SURPRISING PERIODICITY

The first line of evidence that the network is using the algorithm described in Section 3.1 is the surprising periodicity in the activations of the transformer. That is, the output of every part of the network is periodic as a function of the input tokens.

**Periodicity in the embeddings.** We start by examining the embeddings. We apply a Fourier transform along the input dimension of the embedding matrix $W_E$ then compute the $\ell_2$-norm along the other dimension; results are shown in Figure 3. We plot only the components for the first 56 frequencies, as the norm of the components for frequencies $k$ and $P - k$ are symmetric. The embedding matrix $W_E$ is sparse in the Fourier basis–it only has significant nonnegligible norm at 6 frequencies. Of these frequencies, only 5 appear to be used significantly in later parts of the model (corresponding to $k \in \{14, 35, 41, 42, 52\}$). We dub these the *key frequencies* of the model.

**Periodicity in attention heads and MLP neuron activations.** This periodic structure recurs throughout the network. As an example, we plot the attention weight at position 0 for every combination of two inputs for head 0 in Figure 4. The attention exhibits a periodic structure with frequency

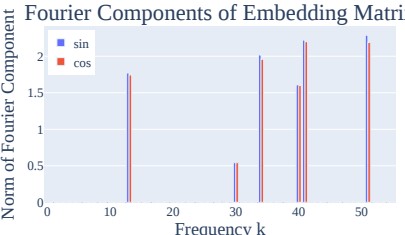 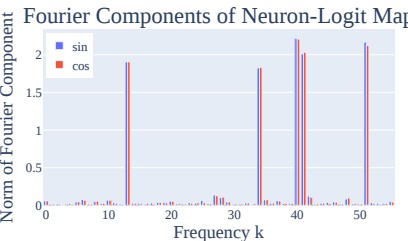

Figure 3: *(Left)* The norms of the Fourier components in the embedding matrix $W_E$. As discussed in Section 4.1, the sparsity of $W_E$ in the Fourier basis is evidence that the network is operating in this basis. Of the six non-zero frequencies, five "key frequencies" appear in later parts of the network, corresponding to $k \in \{14, 35, 41, 42, 52\}$. *(Right)* Norm of Fourier components of the neuron-logit map $W_L$. A Fourier transform is taken over the logit axis, and then the norm is taken over the neuron axis. As discussed in Section 4.2, $W_L$ is well-approximated by the 5 key frequencies $w_k$.

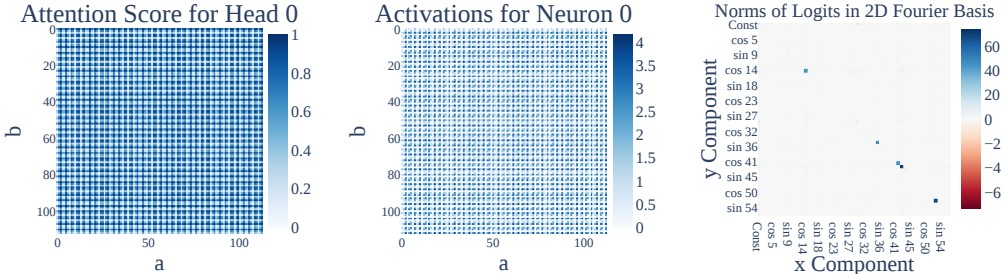

Figure 4: *(Left)* The attention score for head 0 from the token '=' to '$a$', as a function of inputs $a, b$. *(Center)* The activations of MLP neuron 0 given inputs $a, b$. Both the attention scores and the neuron activations are periodic (Section 4.1). *(Right)* The norm of the Fourier components of the logits (2D Fourier transform is taken over the inputs $a, b$, and then norm is taken over the logit axis). There are 20 significant components corresponding to the 5 key frequencies (Section 4.1).

$k = 35$. In Figure 4, we also plot the activations of MLP neuron 0 for every combination of inputs. The activations are periodic with frequency $k = 42$. We see similar patterns for other attention heads and MLP neurons (Appendix C.1).

**Periodicity in logits.** Finally, the logits are also periodic. In Figure 4, we represent the logits in the 2D Fourier basis over the inputs, then take the $\ell_2$-norm over the output dimension. There are only twenty components with significant norm, corresponding to the products of sines and cosines for the five key frequencies $w_k$. These show up as five $2 \times 2$ blocks in Figure 4.

## 4.2 MECHANISTIC EVIDENCE: COMPOSING MODEL WEIGHTS

We now demonstrate that the model implements the trigonometric identity (1) as follows: the functions $\cos(w_k(a+b))$, $\sin(w_k(a+b))$ are linearly represented in the MLP activations, and the unembed matrix reads these linear directions and multiplies them by $\cos(w_k c)$, $\sin(w_k c)$ respectively.

We will do this in two steps. First, we show that $W_L$ (the matrix mapping MLP activations to logits) is (approximately) rank 10 and can be well approximated as:

$$W_L = \sum_{k \in \{14, 35, 41, 42, 52\}} \cos(w_k) u_k^T + \sin(w_k) v_k^T \tag{2}$$

for some $u_k, v_k \in \mathbb{R}^{512}$, where $\cos(w_k), \sin(w_k) \in \mathbb{R}^{113}$ are vectors whose $c$th entry is $\cos(w_k c)$ and $\sin(w_k c)$. Second, note that our model implements the logits for $a, b$ as:

$$\text{Logits}(a, b) = W_L \text{MLP}(a, b) \approx \sum_k \cos(w_k) u_k^T \text{MLP}(a, b) + \sin(w_k) v_k^T \text{MLP}(a, b) \tag{3}$$

We check empirically that the terms $u_k^T \text{MLP}(a, b)$ and $v_k^T \text{MLP}(a, b)$ are approximate multiples of $\cos(w_k(a+b))$ and $\sin(w_k(a+b))$ ($> 90\%$ of variance explained). Thus the network computes trigonometric functions in the MLP and reads them off as claimed. As a sanity check, we confirm

| $W_L$ Component | Fourier components of $u_k^T \text{MLP}(a,b)$ or $v_k^T \text{MLP}(a,b)$ | FVE |
|---|---|---|
| $\cos(w_{14}c)$ | $44.6\cos(w_{14}a)\cos(w_{14}b) - 43.6\sin(w_{14}a)\sin(w_{14}b) \approx 44.1\cos(w_{14}(a+b))$ | 93.2% |
| $\sin(w_{14}c)$ | $44.1\sin(w_{14}a)\cos(w_{14}b) + 44.1\cos(w_{14}a)\sin(w_{14}b) \approx 44.1\sin(w_{14}(a+b))$ | 93.5% |
| $\cos(w_{35}c)$ | $40.7\cos(w_{35}a)\cos(w_{35}b) - 43.6\sin(w_{35}a)\sin(w_{35}b) \approx 42.2\cos(w_{35}(a+b))$ | 96.8% |
| $\sin(w_{35}c)$ | $41.8\sin(w_{35}a)\cos(w_{35}b) + 41.8\cos(w_{35}a)\sin(w_{35}b) \approx 41.8\sin(w_{35}(a+b))$ | 96.5% |
| $\cos(w_{41}c)$ | $44.8\cos(w_{41}a)\cos(w_{41}b) - 44.8\sin(w_{41}a)\sin(w_{41}b) \approx 44.8\cos(w_{41}(a+b))$ | 97.0% |
| $\sin(w_{41}c)$ | $44.5\sin(w_{41}a)\cos(w_{41}b) + 44.5\cos(w_{41}a)\sin(w_{41}b) \approx 44.5\sin(w_{41}(a+b))$ | 97.0% |
| $\cos(w_{42}c)$ | $64.6\cos(w_{42}a)\cos(w_{42}b) - 68.5\sin(w_{42}a)\sin(w_{42}b) \approx 66.6\cos(w_{42}(a+b))$ | 96.4% |
| $\sin(w_{42}c)$ | $67.8\sin(w_{42}a)\cos(w_{42}b) + 67.8\cos(w_{42}a)\sin(w_{42}b) \approx 67.8\sin(w_{42}(a+b))$ | 96.4% |
| $\cos(w_{52}c)$ | $60.5\cos(w_{52}a)\cos(w_{52}b) - 65.5\sin(w_{52}a)\sin(w_{52}b) \approx 63.0\cos(w_{52}(a+b))$ | 97.4% |
| $\sin(w_{52}c)$ | $64.5\sin(w_{52}a)\cos(w_{52}b) + 64.5\cos(w_{52}a)\sin(w_{52}b) \approx 64.5\sin(w_{52}(a+b))$ | 98.2% |

Table 1: For each of the directions $u_k$ or $v_k$ (corresponding to the $\cos(w_k)$ and $\sin(w_k)$ components respectively) in the unembedding matrix, we take the dot product of the MLP activations with that direction, then perform a Fourier transform (middle column; only two largest coefficients shown). We then compute the fraction of variance explained (FVE) if we replace the projection with a single term proportional to $\cos(w_k(a+b))$ or $\sin(w_k(a+b))$, and find that it is consistently close to 1.

that the logits are indeed well-approximated by terms of the form $\cos(w_k(a+b-c))$ (95% of variance explained).

$W_L$ **is well approximated by** $\cos(w_k c)$ **and** $\sin(w_k c)$**.** We perform a discrete Fourier transform (DFT) on the logit axis of $W_L$ and look at the 10 directions $u_k, v_k$ corresponding to $\sin(w_k)$ and $\cos(w_k)$. When we approximate $W_L$ with $\sum_{k \in \{14,35,41,42,52\}} \cos(w_k) u_k^T + \sin(w_k) v_k^T$, the residual has Frobenius norm that is under $0.55\%$ of the norm of $W_L$. This shows that $W_L$ is well approximated by the 10 directions corresponding to $\cos(w_k)$ and $\sin(w_k)$ for each of the five key frequencies. We also plot the norms of each direction in Figure 3, and find that no Fourier component outside the 5 key frequencies has significant norm.

**The unembedding matrix "reads off" terms of the form** $\cos(w_k(a+b))$ **and** $\sin(w_k(a+b))$ **from the MLP neurons.** Next, we take the dot product of the MLP activations with each of the directions $u_k, v_k$ for $k \in \{14, 35, 41, 42, 52\}$. Table 1 displays the results: the dot products $u_k^T \text{MLP}(a,b)$ and $v_k^T \text{MLP}(a,b)$ are well approximated by a multiple of terms of the form

$$\cos(w_k(a+b)) = \cos(w_k a)\cos(w_k b) - \sin(w_k a)\sin(w_k b), \text{ and}$$
$$\sin(w_k(a+b)) = \sin(w_k a)\cos(w_k b) + \cos(w_k a)\sin(w_k b).$$

That is, for each key frequency $k$, $u_k$ and $v_k$ are linear directions in the space of MLP neuron activations that represent $\cos(w_k(a+b))$ and $\sin(w_k(a+b))$.

**Logits are well approximated by a weighted sum of** $\cos(w_k(a+b-c))$**s.** We approximate the output logits as the sum $\sum_k \alpha_k \cos(w_k(a+b-c))$ for $k \in \{14, 35, 41, 42, 52\}$ and fit the coefficients $\alpha_k$ via ordinary least squares. This approximation explains 95% of the variance in the original logits. This is surprising—the output logits are a $113 \cdot 113 \cdot 113$ dimensional vector, but are well-approximated with just the 5 directions predicted by our interpretation. If we evaluate test loss using this logit approximation, we actually see an *improvement* in loss, from $2.4 \cdot 10^{-7}$ to $4.7 \cdot 10^{-8}$.

Taken together, these results confirm that the model computes sums of terms of the form $\cos(w_k(a+b-c)) = \cos(w_k(a+b))\cos(w_k c) + \sin(w_k(a+b))\sin(w_k c)$.

### 4.3 ZOOMING IN: APPROXIMATING NEURONS WITH SINES AND COSINES

In the previous section, we showed how the model computes its final logits by using $W_L$ to "read off" trigonometric identities represented in the MLP neurons. We now examine the attention heads and MLP neurons to understand how the identities come to be represented at the MLP layer. In Appendix C.1.2, we show that two of the attention heads approximately compute degree-2 polynomials of sines and cosines of a particular frequency (and the other two are used to increase the magnitude of the input embeddings in the residual stream). Here, we show that most neurons are also well-approximated by degree-2 polynomials, and the map from neurons to logits is localized by frequency.

**Most MLP neurons approximately compute a degree-2 polynomial of a single frequency.** We next try to approximate the activations of each MLP neuron by a degree-2 polynomial of one of the 5 key frequencies. As shown in Figure 5, out of 512 total neurons, 433 ($84.6\%$) have over $85\%$ of their variance explained with a single frequency.

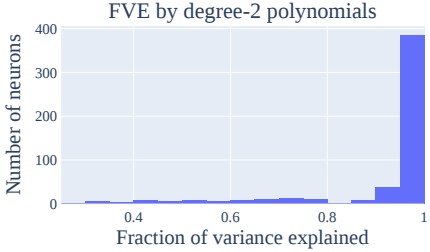 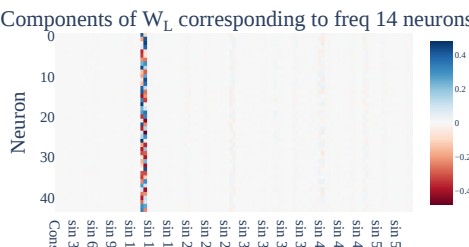

Figure 5: (Left) Most neurons are well-approximated by degree-2 polynomials of a single frequency. (Right) A heatmap showing weights in $W_L$ corresponding to each of the 44 neurons of frequency 14. The non-trivial components correspond to $\sin(w_k)$ and $\cos(w_k)$ for $k = 14$.

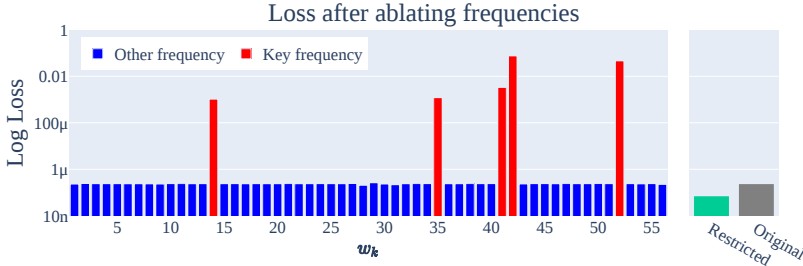

Figure 6: The loss of the transformer (lower=better) when ablating each frequency $k \in \{1, 2, ..., 56\}$ and *everything except for* the five key frequencies (restricted loss). We include the original unablated loss for reference. Ablating key frequencies causes a performance drop, while the other ablations do not harm performance.

**Maps to the logits are localized by frequency.** We partition these 433 neurons by the frequencies with the highest variance explained. For each resulting subset, the map $W_L$ from neurons to logits has only two non-trivial components, corresponding to sine and cosine at that frequency. For example, in Figure 5 we plot the 44 columns of $W_L$ corresponding to the 44 neurons in the $k = 14$ cluster and find that the only non-negligible components are $\sin\left(\frac{2k\pi}{P}\right)$ and $\cos\left(\frac{2k\pi}{P}\right)$ for $k = 14$.

### 4.4 CORRECTNESS CHECKS: ABLATIONS

In previous sections, we showed that various components of the model were well-approximated by sparse combinations of sines and cosines. We verify that these approximations are faithful to the model's functionality, by replacing each component with its approximation. This generally does not hurt the performance of the model and in some cases *improves* it.

**MLP neurons.** In Section 4.3, we identified 433 neurons that were well-approximated by a degree-2 polynomial. We replace each of these neurons' activation value by the corresponding polynomial, leaving the other neurons untouched. This increases loss by only 3% in relative terms (from $2.41 \cdot 10^{-7}$ to $2.48 \cdot 10^{-7}$) and has no effect on accuracy.

We can instead apply a stricter ablation to the MLP layer and restrict each neuron's activation to just the components of the polynomial corresponding to terms of the form $\cos(w_k(a + b))$ and $\sin(w_k(a + b))$ in the key frequencies. This *improves* loss by 77% (to $5.54 \cdot 10^{-8}$), validating that the logits are calculated by trig identities of neurons as detailed in Section 4.2.

**Logit frequencies.** Next, we ablate various components of the final logits in the Fourier space. To do so, we take a 2D DFT on the $113 \cdot 113 \cdot 113$ logit matrix over all $113 \cdot 113$ pairs of inputs to get the logits in the Fourier basis, then set various frequencies in this basis to 0.

We begin by ablating the components corresponding to each of the key frequencies. As reported in Figure 6, ablating any key frequency causes a significant increase in loss. This confirms that the five frequencies identified in previous sections are indeed necessary components of the transformer. In contrast, ablating other frequencies does not hurt the model at all.

We then ablate *all* $113 \cdot 113 - 40$ of the Fourier components besides key frequencies; this ablation actually *improves* performance (loss drops 70% to $7.24 \cdot 10^{-8}$).

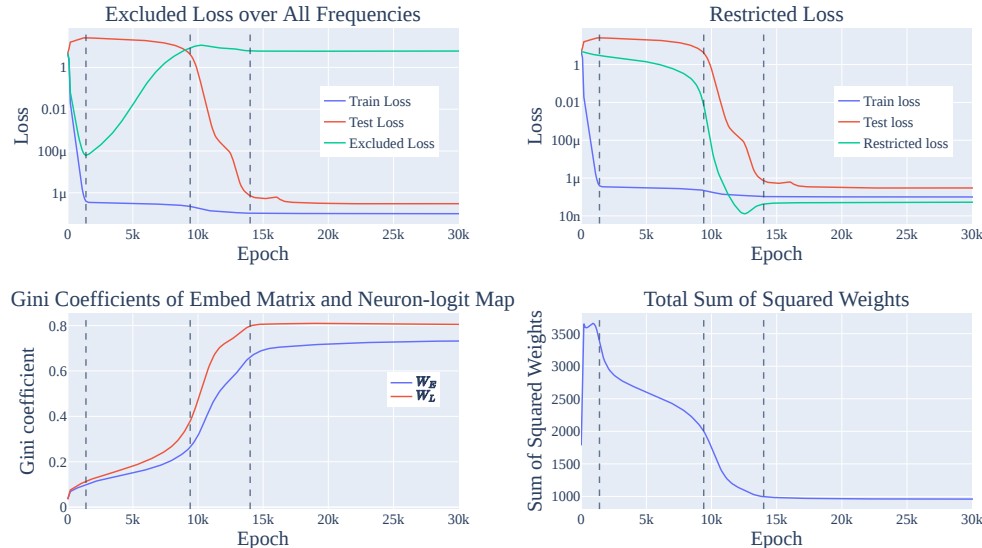

Figure 7: How each of the progress measures in Section 5.1 changes over the course of training. The lines delineate the 3 phases of training: memorization, circuit formation, and cleanup (and a final stable phase). *(Top Left)* Excluded loss increases during circuit formation, while train and test loss remain flat. *(Top Right)* The restricted loss begins declining before test loss declines, but has an inflection point when grokking begins to occur. *(Bottom Left)* The Gini coefficient of the norms of the Fourier components of $W_E$ and $W_L$ increase sharply during cleanup. *(Bottom Right)* The sums of squared weights decreases smoothly during circuit formation and more sharply during cleanup, indicating that both phases are linked to weight decay.

**Directions in $W_L$.** In Section 4.2, we found that $W_L$ is well approximated by the 10 directions corresponding to the cosine and sine of key frequencies. If we project the MLP activations to these 10 directions, loss decreases 50% to $1.19 \cdot 10^{-7}$. If we instead projected the MLP activations onto the nullspace of these 10 directions, loss increases to $5.27$—worse than uniform. This suggests that the network achieves low loss using these and *only* these 10 directions.

## 5 Understanding grokking behavior using progress measures

We now use our mechanistic understanding of the network to define two *progress measures*: metrics that can be computed during training that track the progress of the model over the course of training, including during phase transitions. This allows us to study *how* the network reaches its final solution.

### 5.1 Progress measures

We translate the ablations in Section 4.4 into two progress measures: restricted and excluded loss.

**Restricted loss.** Since the final network uses a sparse set of frequencies $w_k$, it makes sense to check how well intermediate versions of the model can do using only those frequencies. To measure this, we perform a 2D DFT on the logits to write them as a linear combination of waves in $a$ and $b$, and set all terms besides the constant term and the 20 terms corresponding to $\cos(w_k(a+b))$ and $\sin(w_k(a+b))$ for the five key frequencies to 0. We then measure the loss of the ablated network.

**Excluded loss.** Instead of keeping the important frequencies $w_k$, we next remove *only* those key frequencies from the logits but keep the rest. We measure this on the *training* data to track how much of the performance comes from Fourier multiplication versus memorization. The idea is that the memorizing solution should be spread out in the Fourier domain, so that ablating a few directions will leave it mostly unaffected, while the generalizing solution will be hurt significantly.

Beyond these, we will also measure (1) the Gini coefficient (Hurley & Rickard, 2009) of the norms of the Fourier components of $W_E$ and $W_L$, which measures the sparsity of $W_E$ and $W_L$ in the Fourier basis, and (2) the $\ell_2$-norm of the weights during training, since weight decay should push these down once the train loss is near zero.

## 5.2 Phases of grokking: memorization, circuit formation, and cleanup

Using the mainline model from Section 4, we plot the excluded loss, restricted loss, Gini coefficient of the matrices $W_U$ and $W_L$, and sum of squared weights in Figure 7. We find that training splits into three phases, which we call the memorization, circuit formation, and cleanup phases. (We show similar results for other models in Appendix C.2.)

**Memorization** (Epochs 0k–1.4k). We first observe a decline of both excluded and train loss, with test and restricted loss both remaining high and the Gini coefficient staying relatively flat. In other words, the model memorizes the data, and the frequencies $w_k$ used by the final model are unused.

**Circuit formation** (Epochs 1.4k–9.4k). In this phase, excluded loss rises, sum of squared weights falls, restricted loss starts to fall, and test and train loss stay flat. This suggests that the model's behavior on the train set transitions smoothly from the memorizing solution to the Fourier multiplication algorithm. The fall in the sum of squared weights suggests that circuit formation likely happens due to weight decay. Notably, the circuit is formed *well before* grokking occurs.

**Cleanup** (Epochs 9.4k–14k). In this phase, excluded loss plateaus, restricted loss continues to drop, test loss suddenly drops, and sum of squared weights sharply drops. As the completed Fourier multiplication circuit both solves the task well and has lower weight than the memorization circuit, weight decay encourages the network to shed the memorized solution in favor of focusing on the Fourier multiplication circuit. This is most cleanly shown in the sharp increase in the Gini coefficient for the matices $W_E$ and $W_L$, which shows that the network is becoming sparser in the Fourier basis.

## 5.3 Grokking and Weight Decay

In the previous section, we saw that each phase of grokking corresponded to an inflection point in the $\ell_2$-norm of the weights. This suggests that weight decay is an important component of grokking and drives progress towards the generalizing solution. In Appendix D.1, we provide additional evidence that weight decay is necessary for grokking: smaller amounts of weight decay causes the network to take significantly longer to grok (echoing the results on toy models from Liu et al. (2022)), and our networks do not grok on the modular arithmetic task without weight decay or some other form of regularization. In Appendix C.2, we also find that the amount of data affects grokking: when networks are provided with enough data, there is no longer a gap between the train and test losses (instead, both decline sharply some number of epochs into training). Finally, in Appendix D.3 we replicate these results on several additional algorithmic tasks.

## 6 Conclusion and discussion

In this work, we use mechanistic interpretability to define progress measures for small transformers trained on a modular addition task. We find that the transformers embed the input onto rotations in $\mathbb{R}^2$ and compose the rotations using trigonometric identities to compute $a + b \mod 113$. Using our reverse-engineered algorithm, we define two progress measures, along which the network makes continuous progress toward the final algorithm prior to the grokking phase change. We see this work as a proof of concept for using mechanistic interpretability to understand emergent behavior.

**Larger models and realistic tasks**. In this work, we studied the behavior of small transformers on a simple algorithmic task, solved with a single circuit. On the other hand, larger models use larger, more numerous circuits to solve significantly harder tasks (Cammarata et al., 2020; Wang et al., 2022). The analysis reported in this work required significant amounts of manual effort, and our progress metrics are specific to small networks on one particular algorithmic task. Methods for automating the analysis and finding task-independent progress measures seem necessary to scale to other, larger models. We discuss possible scenarios for more realistic applications in Appendix F.

**Discovering phase change thresholds**. While the progress measures we defined in Section 5.1 increase relatively smoothly before the phase transition (and suffice to allow us to understand grokking for this task) we lack a general notion of criticality that would allow us to predict *when* the phase transition will happen ex ante. Future work should develop theory and practice in order to apply progress measures to predict the timing of emergent behavior.

## REPRODUCIBILITY STATEMENT

An annotated Colab notebook containing the code to replicate our results, including download instructions for model checkpoints, is available at https://bit.ly/grokking-progress-measures-website.

## AUTHOR CONTRIBUTIONS

**Neel Nanda** was the primary research contributor. He reverse engineered the weights of the mainline model to discover the Fourier multiplication algorithm and found the lines of evidence in Section 4. He also discovered the restricted and excluded loss progress measures and that grokking in mainline model could be divided into three discrete phases. Finally, he found the link between grokking, limited data, and phase transitions by exhibiting grokking in other settings with phase transitions.

**Lawrence Chan** was invaluable to the framing and technical writing of this work. In addition, he created the Gini coefficient progress measure and performed the analysis in the appendices exploring to what extent the results on the mainline model applied to the other small transformer models, including with other random seeds, architectures, prime moduli, and regularization methods.

**Tom Lieberum** contributed to the early stages of this work by creating a minimal setup of grokking with a 1L Transformer on the modular addition task with no LayerNorm and finding the surprising periodicity within the model's internals.

**Jess Smith** performed experiments exploring grokking with different random seeds, architectures, and other hyper-parameters.

**Jacob Steinhardt** helped clarify and distill the results, provided significant amounts of editing and writing feedback, and suggested the progress measure frame.

## ACKNOWLEDGMENTS

In writing this paper, our thinking and exposition was greatly clarified by correspondence with and feedback from Oliver Balfour, David Bau, Sid Black, Nick Cammarata, Stephen Casper, Bilal Chughtai, Arthur Conmy, Xander Davies, Ben Edelman, Nelson Elhage, Ryan Greenblatt, Jacob Hilton, Evan Hubinger, Zac Kenton, Janos Kramar, Lauro Langosco, Tao Lin, David Lindner, Eric Michaud, Vlad Mikulik, Noa Nabeshima, Chris Olah, Michela Paganini, Michela Paganini, Alex Ray, Rohin Shah, Buck Shlegeris, Alex Silverstein, Ben Toner, Johannes Treutlein, Nicholas Turner, Vikrant Varma, Vikrant Varma, Kevin Wang, Martin Wattenberg, John Wentworth, and Jeff Wu.

We'd also like to thank Adam Gleave and Chengcheng Tan for providing substantial editing help, and Noa Nabeshima and Vlad Mikulik for pair programming with Neel.

This work draws heavily on the interpretability techniques and framework developed by Elhage et al. (2021) and Olsson et al. (2022).

We trained our models using PyTorch (Paszke et al., 2019) and performed our data analysis using NumPy (Harris et al., 2020), Pandas (Wes McKinney, 2010), and einops (Rogozhnikov, 2022). Our figures were made using Plotly (Plotly Technologies Inc., 2015).

Neel would like to thank Jemima Jones for providing practical and emotional support as he navigated personal challenges while contributing to this paper, and to the Schelling Residency for providing an excellent research environment during the distillation stage. He would also like to thank the Anthropic interpretability team, most notably Chris Olah, for an incredibly generous amount of mentorship during his time there, without which this investigation would never have happened.

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

## A  MATHEMATICAL STRUCTURE OF THE TRANSFORMER

We follow the conventions and notation of Elhage et al. (2021) in describing our model. Here, we briefly recap their notation and examine it in our specific case.

We denote our hyperparameters as follows: $d_{vocab} = 113$ is the size of the input and output spaces (treating '=' separately), $d_{model} = 128$ is the width of the residual stream (i.e. embedding size), $d_{head} = 32$ is the size of query, key and value vectors for a single attention head, and $d_{mlp} = 512$ is the number of neurons.

We denote the parameters as follows: $W_E$ (embedding layer); $W_{pos}$ (positional embedding); $W_Q^j$ (queries), $W_K^j$ (keys), $W_V^j$ (values), $W_O^j$ (attention output) (the 4 weight matrices of head $j$ in the attention layer); $W_{in}$ and $b_{in}$ for the input linear map of the MLP layer; $W_{out}$ and $b_{out}$ for the output linear map of the MLP layer; and $W_U$ (unembedding layer). Note that we do not have biases in our embedding, attention layer or unembedding, and we do not tie the matrices for the embedding/unembedding layers.

We now describe the mathematical structure of our network. Note that loss is only calculated from the logits on the final token, and information only moves between tokens during the attention layer, so our variables from the end of the attention layer onwards only refer to the final token. We use $t_i$ to denote the token in position $i$ (as a one-hot encoded vector), $p_i$ to denote the $i$th positional embedding, $x_i^{(0)}$ to denote the initial residual stream on token with index $i$, $A^{(i)}$ to denote the attention scores from = to all previous tokens from head $i$, $x^{(1)}$ to denote the residual stream after the attention layer on the final token, MLP to denote the neuron activations in the MLP layer on the final token, $x^{(2)}$ the final residual stream on the final token, Logits the logits on the final token.

The logits are calculated via the following equations:

$$x_i^{(0)} = W_E t_i + p_i$$
$$A^j = \text{softmax}(x^{(0)^T} W_K^{j^T} W_Q^j x_2^{(0)})$$
$$x^{(1)} = [\sum_j W_O^j W_V^j (x^{(0)} \cdot A^j)] + x_2^{(0)}$$
$$\text{MLP} = \text{ReLU}(W_{in} x^{(1)})$$
$$x^{(2)} = W_{out} N + x^{(1)} = W_{out} \text{ReLU}(W_{in} x^{(1)}) + x^{(1)}$$
$$\text{Logits} = W_U x^{(2)}$$

As in Elhage et al. (2021), we refer to the term $W_O^j W_V^j (x^{(0)})$ as the OV circuit for head $j$.

### A.1  EMPIRICAL MODEL SIMPLIFICATIONS

We make two empirical observations:

- The attention paid from '=' to itself is trivial. In practice, the average attention paid is 0.1% to 0.4% for each head, and ablating this does not affect model performance at all.
- The skip connection around the MLP layer is not important for the model's computation and can be ignored. Concretely, if we set it to zero or to its average (zero or mean ablation) then model accuracy is unchanged, and loss goes from $2.4 \cdot 10^{-7}$ to $9.12 \cdot 10^{-7}$ and $7.25 \cdot 10^{-7}$ respectively. This is a significant increase in loss, but from such a small baseline that we can still ignore it and reverse engineer the model's computation. (That being said, both the attention heads and the skip connection around them are crucial to the functioning of the model: zero ablating attention heads increases loss to 24.3, while zero ablating the skip connection around the attention heads increases loss to 19.1, both *significantly* worse than chance.)

A consequence of the first observation is that the attention is now a softmax over 2 elements, i.e. a sigmoid over the difference. And $x_2^{(0)}$ is constant, as it is independent of $x$ and $y$, and the embedding

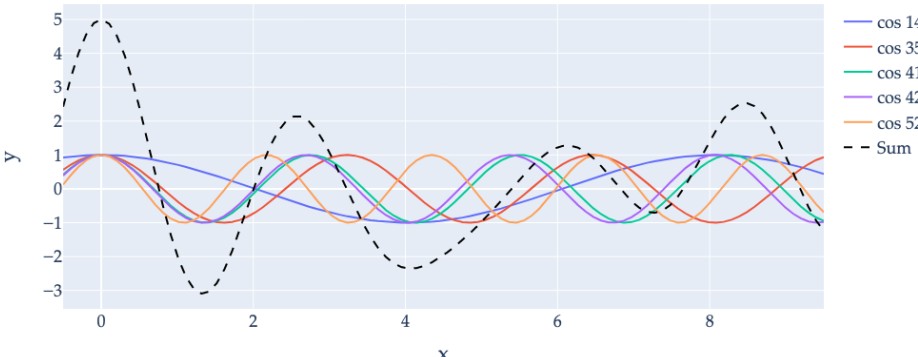

Figure 8: As discussed in Appendix B, while for every $k \in [0, ...P - 1]$, $\cos\left(\frac{2k\pi}{P}x\right)$ achieves its maximum value (1) at $x = 0 \mod 113$, it still has additional peaks at different values that are close to the maximum value. However, by adding together cosine waves of the 5 keyfrequencies, the model constructs a periodic function where the value at $x = 0 \mod 113$ is significantly larger than its value anywhere else.

and positional embedding of '=' are fixed. So $A_0^j = \sigma\left(x_2^{(0)^T} W_Q^{j^T} W_K (x_0^{(0)} - x_1^{(0)})\right)$ (and $A_1^j = 1 - A_0^j$)

A consequence of the second observation is that Logits $\approx W_U W_{out}$MLP, which we denote as $W_L = W_U W_{out}$. From the perspective of the network, $W_L$ is the meaningful matrix, not either of its constituents, since they compose linearly.

## B WHY USE CONSTRUCTIVE INTEREFERENCE?

As demonstrated in Section 4 and Appendix C.2.1, small transformers trained on this task use several different frequencies which they add together. The reason for this is to end up with a function whose value at $x = 0 \mod 113$ is significantly larger than any other $x$.

For example, consider the function $f_{14}(x) = \cos\left(\frac{2\pi \cdot 14}{113}x\right)$. This function has period 113 and is maximized at $x = 0 \mod 113$. However, other values of $x$ cause this function to be close to 1: $f_{14}(8) = f_{14}(105) = 0.998$, $f_{14}(16) = f_{14}(89) = 0.994$, etc.

Now consider $f_{35}(x) = \cos\left(\frac{2\pi \cdot 35}{113}x\right)$. While this function also has period 113 and is maximized at $x = 0 \mod 113$, it turns out that $f_{35}(8) = f_{35}(105) = -0.990$. This means that by adding together $f_{14}$ and $f_{35}$, we end up with a function that is not close to 1 at $x = 8 \mod 113$. Similarly, while $f_{35}(16) = 0.961$, $f_{52}(16) = -0.56$, and so adding a third frequency reduces the peak at $x = 16 \mod 113$.

We show the constructive interference resulting from the cosine waves for the five frequencies used by the mainline model in Figure 8.

## C SUPPORTING EVIDENCE FOR MECHANISTIC ANALYSIS OF MODULAR ARITHMETIC NETWORKS

### C.1 FURTHER ANALYSIS OF THE SPECIFIC TRAINING RUN DISCUSSED IN THE PAPER

In this section, we provide additional evidence relating to the mainline model.

Figure 9: Attention patterns for each head, from the '=' token at the third sequence position to the $a$ token at the first sequence position, as a heatmap over the inputs. All four attention heads exhibit striking periodicity.

| Head | $k$ | $\alpha^j$ | $\beta^j$ | FVE |
|------|-----|------------|-----------|--------|
| 0 | 35 | $-0.26$ | $-0.14$ | 99.03% |
| 1 | 42 | 0.27 | $-0.04$ | 98.49% |
| 2 | 52 | 0.29 | $-0.05$ | 99.07% |
| 3 | 42 | $-0.26$ | 0.04 | 97.91% |

Table 2: For each attention head, we show the pattern from '=' to $a$ is well approximated by $0.5 + \alpha(\cos(w_k a) - \cos(w_k b)) + \beta(\sin(w_k a) - \sin(w_k b))$ and give the coefficients and fraction of variance explained for this approximation.

### C.1.1    PERIODICITY IN THE ACTIVATIONS OF OTHER ATTENTION HEADS

In Figure 9 we plot the attention patterns from the final token '=' to the first token $a$ for all 4 attention heads, as a heatmap over the inputs $a$ and $b$, as this is a scalar for each head. We observe a striking periodicity and further that heads 1 and 3 represent the same frequency while heads 0 and 2 are different.

As shown in Appendix A.1, the attention paid from '=' to itself is negligible, so $A_0^j = 1 - A_1^j$ and it suffices to plot attention to $a$.

### C.1.2    APPROXIMATING ATTENTION HEADS WITH SINES AND COSINES

**Attention heads approximately compute degree-$2$ polynomials of a single frequency or are used to amplify $W_E$.** In order to compute terms like $\cos(w_k(a + b))$, the model needs to compute the product of the sine and cosine embeddings output by $W_E$. As the attention heads are approximately bilinear (product of attention weights and OV circuit), they are a natural place to perform this computation. Indeed, for each head, the attention scores' Fourier transform is concentrated on a single frequency $w_k$. For two of the four heads, the corresponding OV circuit is concentrated on that same frequency. Moreover, the softmax mapping the attention scores to attention weights is in a regime where it behaves approximately linearly (and replacing it with a linear function actually improves performance). Thus the attention weights multiply with the OV output to create degree-2 polynomials of the frequency $w_k$, as would be needed for the cosine/sine addition formulas.

For the remaining two heads, their attention scores approximately sum to one and the OV circuits contain all five key frequencies, suggesting that they are used to increase the magnitude of key frequencies in the residual stream. We confirm all of these claims in Appendix C.1.3.

### C.1.3    THE ATTENTION PATTERN WEIGHTS ARE WELL APPROXIMATED BY DIFFERENCES OF SINES AND COSINES OF A SINGLE FREQUENCY.

The periodicity of the attention heads has a striking form—$A_0^j$ is well approximated by $0.5 + \alpha^j(\cos(w_k a) - \cos(w_k b)) + \beta^j(\sin(w_k a) - \sin(w_k b))$, for some frequency $w_k$ and constants $\alpha^j$ and $\beta^j$ (which may differ for each head). Note further that this simplifies to $0.5 + \gamma(\cos(w_k(a + \theta)) - \cos(w_k(b + \theta)))$ for some constants $\gamma$ and $\theta$. We show the coefficients and fraction of variance explained in Table 1

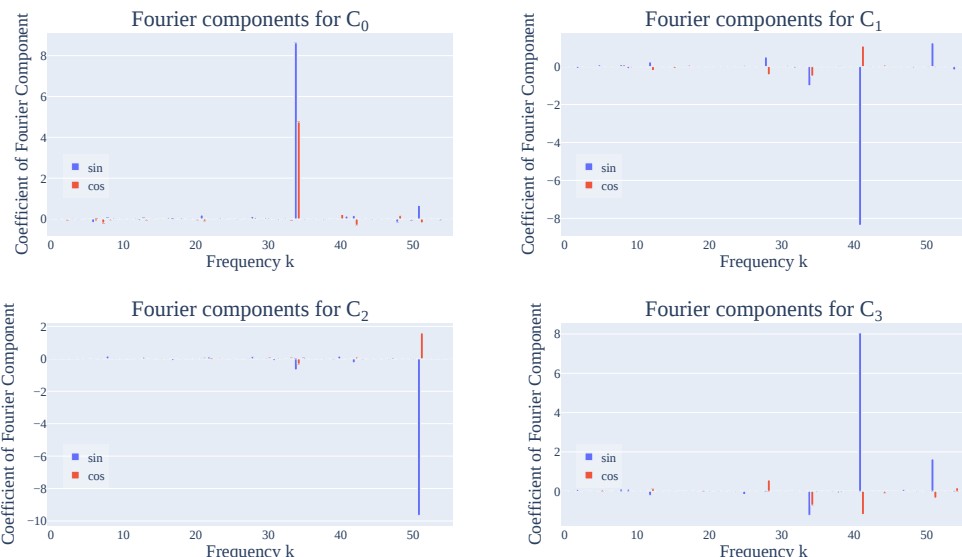

Figure 10: We plot the attention pattern weights $C_j$ in the Fourier basis for each of the four heads $j \in \{0, 1, 2, 3\}$. We observe significant sparsity, with almost all of each term being associated with a single frequency.

**Mechanistic Analysis of Attention Patterns.** We can further mechanistically analyse *how* the model achieves this form. The following is a high-level sketch of what is going on:

First, note that the attention score on position $0$ and head $j$ is just a lookup table on the input token $a$ (of size $P$). To see why, note that $A_0^j = m x_0^{(0)^T} W_K^{j^T} W_Q^j x_2^{(0)}$. $x_2^{(0)}$ is constant since the token is always '=' and $x_0^{(0)} = W_E t_0 + p_0$. So this reduces to $t_0 \cdot C_j + D$ for some constant vector $C_j = W_E^T W_K^{j^T} W_Q^j x_2^{(0)} \in \mathbb{R}^p$ and some scalar $D = p_0^T W_K^{j^T} W_Q^j x_2^{(0)}$. As $t_0$ is one-hot encoded, this is just a lookup table, which we may instead denote as $C_j[a]$

Next, note that the attention pattern from $=\rightarrow 0$ is $\sigma(C_j[a] - C_j[b])$. As argued in Appendix A.1, the attention paid $=\rightarrow=$ is negligible and can be ignored. So the softmax reduces to a softmax over two elements, which is a sigmoid on their difference. As form of $C_j$ does not mention the token index or value, it is the same for position $0$ and $1$.

We now show that $C_j$ is well-approximated by a wave of frequency $w_{k_j}$ for some integer $k_j$. That is, $C_j[a] \approx F_j \cos(w_{k_j} a) + G_j \sin(w_{k_j} a)$. We do this by simply computing $C_j$ and fitting the constants $F_j$ and $G_j$ to minimize $\ell_2$ loss, and display the resulting coefficients for each head in Figure 10. This fit explain $99.02\%$, $95.21\%$, $99.10\%$, $92.42\%$ of the variance of $C_j$ respectively. Interestingly, the coefficients of heads 1 and 3 are almost exactly the opposite of each other.

For each head $j$, $\sigma(C_j[a] - C_j[b]) \approx 0.5 + E_j(C_j[a] - C_j[b])$ for some constant $E_j$—that is, the sigmoid has some linear approximation. (The intercept will be $0.5$ by symmetry.) The striking thing is that, because the inputs to the sigmoid for the attention heads are over a fairly wide range ($[-5, 5]$ roughly), the linear approximation to the sigmoid is a fairly good fit, explaining $97.5\%$ of the variance.

We validate that this is all that is going on, by replacing the sigmoid with the best linear fit. This *improves* performance, decreasing test loss from $2.41 \cdot 10^{-7}$ to $2.12 \cdot 10^{-7}$.

By properties of sinusoidal functions, the attention patterns of each head will be well approximated by $0.5 \pm C_j(\cos(w_{k_j}(a + \theta_j)) - \cos(w_{k_j}(b + \theta_j)))$ - the softmax is linear, with an intercept of $0.5$, and the weights $C_j$ map each token to a score that is a wave in a single frequency. This exactly gives us the periodic form shown in Figure 9.

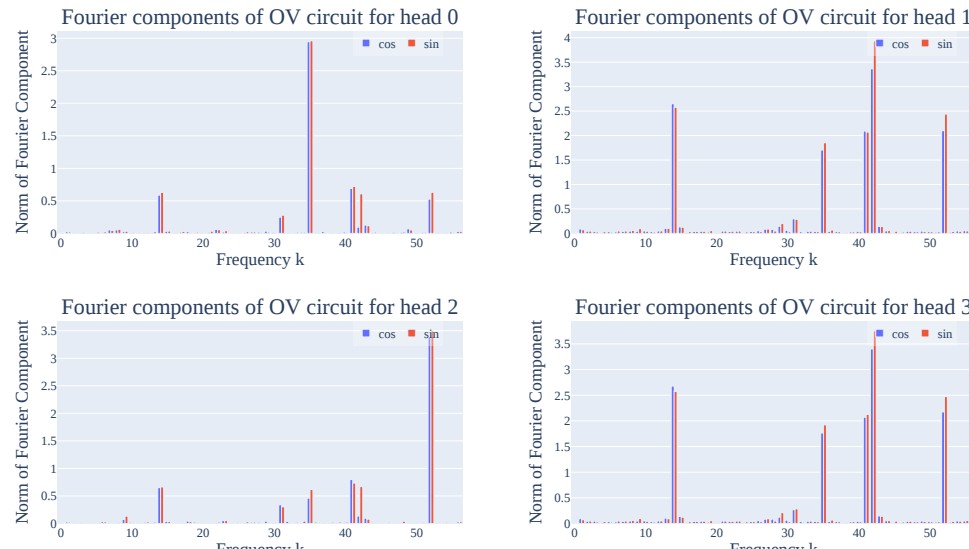

Figure 11: We plot the output of the OV circuit $W_O^j W_V^j x^{(0)}$ in the Fourier basis for each of the four heads $j \in \{0, 1, 2, 3\}$. As with the attention pattern weights $C_j$ in Figure 10, we observe that the only components with significant norm are those corresponding to key frequencies, and that the largest component corresponds to the frequencies of the attention patterns of the attention heads. As attention pattern of heads 1 and 3 are sum to one, but their OV circuits are almost exactly the same and consist of all five key frequencies, this implies that heads 1 and 3 are used to increase the magnitude of key frequencies in the residual stream (Section C.1.3).

Finally, for each head $j$, we plot the output of the OV circuit $W_O^j W_V^j x^{(0)}$ in the Fourier basis and display the results in Figure 11). The largest component of each head corresponding to the frequency of the attention pattern $C_j$, with heads 0 and 2 being almost entirely composed of a sines and cosines of a single frequency. On the other hand, the norms for the components of heads 1 and 3 are almost exactly the same, and contain all five key frequencies. As the coefficients of the attention pattern weights have the opposite non-constant components (Table 2, Figure 10), their attention scores sum almost exactly to 1 across all inputs. This implies that heads 1 and 3 are used to output the first order terms $\sin(w_k), \cos(w_k)$ in the five key frequencies. We speculate that this is because of weight decay encouraging the embeddings $W_E$ to be small, causing the network to allocate two of its attention heads to effectively increasing the size of $W_E$.

Bringing it all together, this implies that attention heads 0 and 2 are approximately computing a degree 2 polynomial of cosines and sines of a single frequency each, while heads 1 and 3 amplify the key frequencies in the residual stream.

### C.1.4 Periodicity in the activations of additional neurons

In Figure 12, we display the activations of four more MLP neurons, as a function of the inputs. As with neuron 0, the activations of these neurons are also periodic in the inputs.

### C.1.5 Additional grokking figures for mainline run

In Figure 13, we display the accuracy of the model when restricting the model to use only the five key frequencies. As with restricted loss, this *improves* model performance during training.

In Figure 14, we show the coefficients of the five key frequencies in the logits, calculated by regressing the logits against the five $\cos(w_k(a + b - c))$ terms.

In Figure 15, we plot the excluded loss if we exclude each of the five key frequencies (as opposed to all five key frequencies).

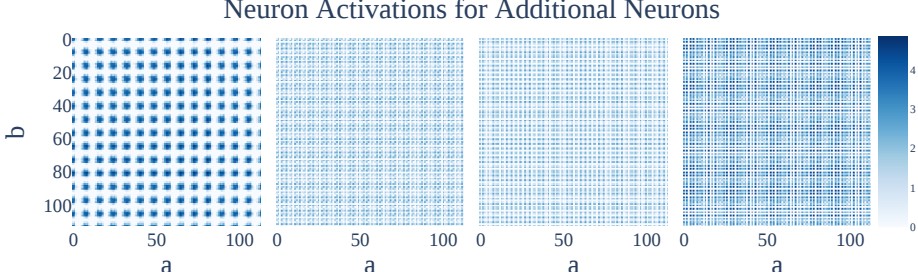

Figure 12: Plots of neuron activations for MLP neurons 1, 2, 3 and 4, for inputs $a, b \in \{0, 1, ..., 112\}$. As with Neuron 0, all of the activation patterns are periodic in both inputs.

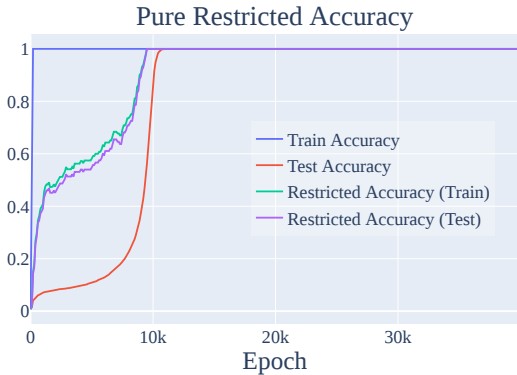

Figure 13: Accuracy when restricting Fourier Components to the five key frequencies. As with restricted loss, this shows that the model figures out how to generalize modulo deleting noise before it removes the noise.

All three of these figures have inflection points corresponding to the relevant phases of grokking, discussed in Section 5.1.

## C.2 ADDITIONAL RESULTS FROM DIFFERENT RUNS

In this section, we plot relevant figures from other runs, either with the same architecture (Appendix C.2.1) or with different architectures or experimental setups (Appendix C.2.2). Note that in general, while all models learn to use variants of the modular arithmetic algorithm, they use a varying number of *different* key frequencies. In order to find the key frequencies to calculate the excluded and restricted loss, we perform a DFT on the neuron-logit map $W_L$, then take the frequencies with nontrivial coefficients.[3]

## C.2.1 ADDITIONAL RESULTS FOR DIFFERENT RUNS WITH THE SAME ARCHITECTURE

In this section, we provide evidence that all 4 other runs (i.e., random seeds) using the experimental setup of our mainline model also use the Fourier multiplication algorithm, and then confirm that the same phases of grokking also occur on these runs.

---

[3]One method for getting a general (model-independent) progress measure for this task is to compute the excluded loss for each of the 56 unique frequencies and then take the max. We omit the plots for this variant of the excluded loss as they are broadly similar.

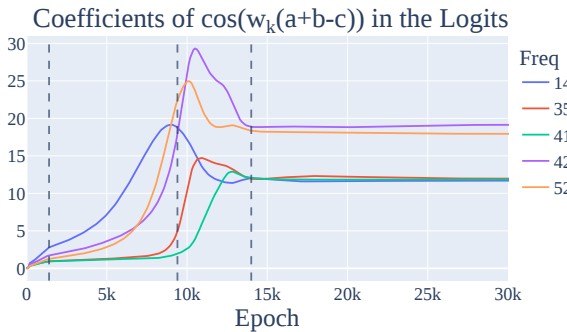

Figure 14: The coefficients of $\cos(w(a+b-c))$ in the logits over the model's training. As with the metrics in the paper, this shows a nice interpolation and growth of each cosine term.

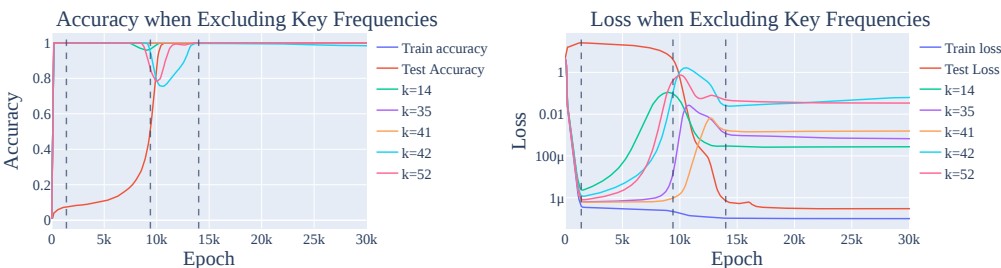

Figure 15: The excluded accuracy (left) and loss (right) if we exclude each of the five key frequencies for our mainline model. As with the excluded loss results in Section 5.1, this shows that the model interpolates between memorising and generalising.

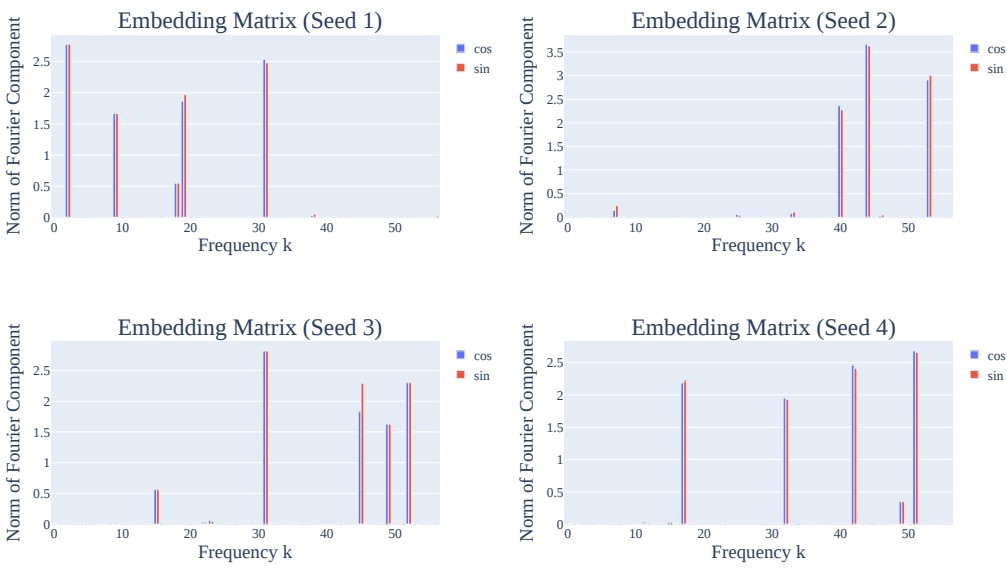

Figure 16: The norms of the Fourier components in the embedding matrix $W_E$ for each of four other random seeds for the original (1 layer) architecture. As discussed in Section 4.1 and Appendix C.2.1, the sparsity of $W_E$ in the Fourier basis is evidence that the network is operating in a Fourier basis.

| $W_L$ Component | Fourier components of $u_k^T \text{MLP}(a,b)$ or $v_k^T \text{MLP}(a,b)$ | FVE |
|---|---|---|
| $\cos(w_2 c)$ | $147.4\cos(w_2 a)\cos(w_2 b) - 145.8\sin(w_2 a)\sin(w_2 b) \approx 146.6\cos(w_2(a+b))$ | 99.2% |
| $\sin(w_2 c)$ | $145.5\cos(w_2 a)\sin(w_2 b) + 145.6\sin(w_2 a)\cos(w_2 b) \approx 145.5\sin(w_2(a+b))$ | 99.1% |
| $\cos(w_9 c)$ | $49.3\cos(w_9 a)\cos(w_9 b) - 48.0\sin(w_9 a)\sin(w_9 b) \approx 48.6\cos(w_9(a+b))$ | 96.4% |
| $\sin(w_9 c)$ | $48.6\cos(w_9 a)\sin(w_9 b) + 48.5\sin(w_9 a)\cos(w_9 b) \approx 48.5\sin(w_9(a+b))$ | 96.7% |
| $\cos(w_{19} c)$ | $58.0\cos(w_{19} a)\cos(w_{19} b) - 58.3\sin(w_{19} a)\sin(w_{19} b) \approx 58.2\cos(w_{19}(a+b))$ | 95.4% |
| $\sin(w_{19} c)$ | $59.3\cos(w_{19} a)\sin(w_{19} b) + 59.4\sin(w_{19} a)\cos(w_{19} b) \approx 59.4\sin(w_{19}(a+b))$ | 93.9% |
| $\cos(w_{31} c)$ | $94.4\cos(w_{31} a)\cos(w_{31} b) - 96.4\sin(w_{31} a)\sin(w_{31} b) \approx 95.4\cos(w_{31}(a+b))$ | 98.4% |
| $\sin(w_{31} c)$ | $97.2\cos(w_{31} a)\sin(w_{31} b) + 97.1\sin(w_{31} a)\cos(w_{31} b) \approx 97.2\sin(w_{31}(a+b))$ | 98.7% |

(a) Seed 1

| $W_L$ Component | Fourier components of $u_k^T \text{MLP}(a,b)$ or $v_k^T \text{MLP}(a,b)$ | FVE |
|---|---|---|
| $\cos(w_{40} c)$ | $97.0\cos(w_{40} a)\cos(w_{40} b) - 99.4\sin(w_{40} a)\sin(w_{40} b) \approx 98.2\cos(w_{40}(a+b))$ | 97.3% |
| $\sin(w_{40} c)$ | $81.3\cos(w_{40} a)\sin(w_{40} b) + 81.3\sin(w_{40} a)\cos(w_{40} b) \approx 81.3\sin(w_{40}(a+b))$ | 92.7% |
| $\cos(w_{44} c)$ | $309.1\cos(w_{44} a)\cos(w_{44} b) - 338.7\sin(w_{44} a)\sin(w_{44} b) \approx 323.9\cos(w_{44}(a+b))$ | 98.5% |
| $\sin(w_{44} c)$ | $327.3\cos(w_{44} a)\sin(w_{44} b) + 327.2\sin(w_{44} a)\cos(w_{44} b) \approx 327.3\sin(w_{44}(a+b))$ | 98.9% |
| $\cos(w_{53} c)$ | $192.1\cos(w_{53} a)\cos(w_{53} b) - 192.2\sin(w_{53} a)\sin(w_{53} b) \approx 192.1\cos(w_{53}(a+b))$ | 97.3% |
| $\sin(w_{53} c)$ | $166.7\cos(w_{53} a)\sin(w_{53} b) + 166.8\sin(w_{53} a)\cos(w_{53} b) \approx 166.8\sin(w_{53}(a+b))$ | 95.7% |

(b) Seed 2

| $W_L$ Component | Fourier components of $u_k^T \text{MLP}(a,b)$ or $v_k^T \text{MLP}(a,b)$ | FVE |
|---|---|---|
| $\cos(w_{31} c)$ | $156.1\cos(w_{31} a)\cos(w_{31} b) - 156.5\sin(w_{31} a)\sin(w_{31} b) \approx 156.3\cos(w_{31}(a+b))$ | 99.3% |
| $\sin(w_{31} c)$ | $150.7\cos(w_{31} a)\sin(w_{31} b) + 150.7\sin(w_{31} a)\cos(w_{31} b) \approx 150.7\sin(w_{31}(a+b))$ | 98.9% |
| $\cos(w_{45} c)$ | $72.5\cos(w_{45} a)\cos(w_{45} b) - 76.8\sin(w_{45} a)\sin(w_{45} b) \approx 74.6\cos(w_{45}(a+b))$ | 95.9% |
| $\sin(w_{45} c)$ | $74.7\cos(w_{45} a)\sin(w_{45} b) + 74.6\sin(w_{45} a)\cos(w_{45} b) \approx 74.6\sin(w_{45}(a+b))$ | 96.6% |
| $\cos(w_{49} c)$ | $45.9\cos(w_{49} a)\cos(w_{49} b) - 45.5\sin(w_{49} a)\sin(w_{49} b) \approx 45.7\cos(w_{49}(a+b))$ | 97.0% |
| $\sin(w_{49} c)$ | $45.8\cos(w_{49} a)\sin(w_{49} b) + 45.8\sin(w_{49} a)\cos(w_{49} b) \approx 45.8\sin(w_{49}(a+b))$ | 96.9% |
| $\cos(w_{52} c)$ | $71.6\cos(w_{52} a)\cos(w_{52} b) - 72.1\sin(w_{52} a)\sin(w_{52} b) \approx 71.9\cos(w_{52}(a+b))$ | 98.5% |
| $\sin(w_{52} c)$ | $68.7\cos(w_{52} a)\sin(w_{52} b) + 68.7\sin(w_{52} a)\cos(w_{52} b) \approx 68.7\sin(w_{52}(a+b))$ | 97.9% |

(c) Seed 3

| $W_L$ Component | Fourier components of $u_k^T \text{MLP}(a,b)$ or $v_k^T \text{MLP}(a,b)$ | FVE |
|---|---|---|
| $\cos(w_{17} c)$ | $66.0\cos(w_{17} a)\cos(w_{17} b) - 63.5\sin(w_{17} a)\sin(w_{17} b) \approx 64.8\cos(w_{17}(a+b))$ | 96.4% |
| $\sin(w_{17} c)$ | $66.4\cos(w_{17} a)\sin(w_{17} b) + 66.4\sin(w_{17} a)\cos(w_{17} b) \approx 66.4\sin(w_{17}(a+b))$ | 94.9% |
| $\cos(w_{32} c)$ | $68.7\cos(w_{32} a)\cos(w_{32} b) - 68.4\sin(w_{32} a)\sin(w_{32} b) \approx 68.5\cos(w_{32}(a+b))$ | 96.2% |
| $\sin(w_{32} c)$ | $68.0\cos(w_{32} a)\sin(w_{32} b) + 68.0\sin(w_{32} a)\cos(w_{32} b) \approx 68.0\sin(w_{32}(a+b))$ | 96.3% |
| $\cos(w_{42} c)$ | $100.4\cos(w_{42} a)\cos(w_{42} b) - 96.0\sin(w_{42} a)\sin(w_{42} b) \approx 98.2\cos(w_{42}(a+b))$ | 97.9% |
| $\sin(w_{42} c)$ | $100.2\cos(w_{42} a)\sin(w_{42} b) + 100.1\sin(w_{42} a)\cos(w_{42} b) \approx 100.1\sin(w_{42}(a+b))$ | 98.6% |
| $\cos(w_{51} c)$ | $118.0\cos(w_{51} a)\cos(w_{51} b) - 116.2\sin(w_{51} a)\sin(w_{51} b) \approx 117.1\cos(w_{51}(a+b))$ | 99.0% |
| $\sin(w_{51} c)$ | $114.3\cos(w_{51} a)\sin(w_{51} b) + 114.2\sin(w_{51} a)\cos(w_{51} b) \approx 114.2\sin(w_{51}(a+b))$ | 98.5% |

(d) Seed 4

Table 3: For each of the directions in the neuron-logit map $W_L$ of the final models from 4 other random seeds (Appendix C.2.1), we project the MLP activations in that direction then perform a Fourier transform. For brevity, we omit terms with coefficients less than 15% of the largest coefficient. We then compute the fraction of variance explained (FVE) if we replace the projection with a multiple of a single term of the form $\cos(w_k(a+b))$ or $\sin(w_k(a+b))$, and find that this is consistently close to 1.

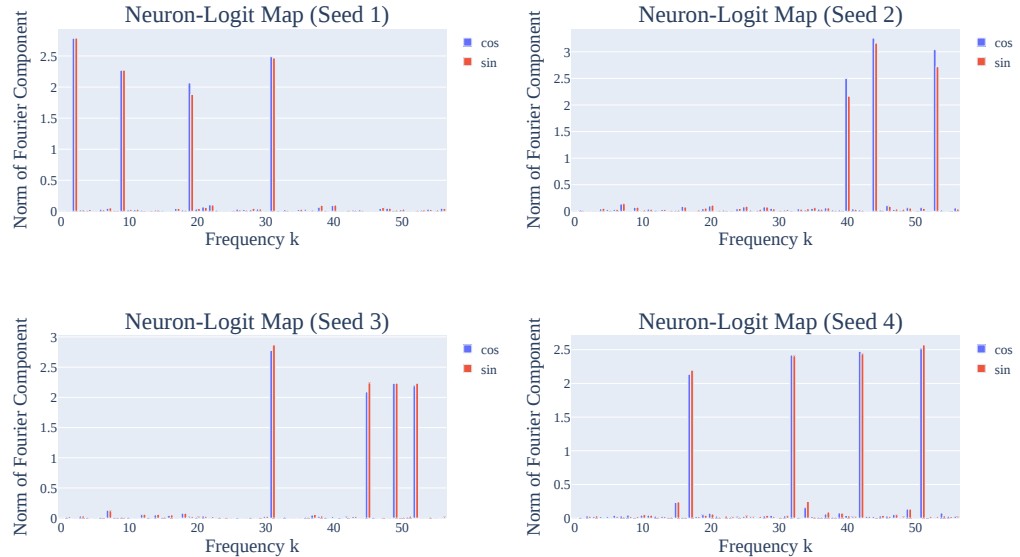

Figure 17: The norms of the direction corresponding to sine and cosine waves in the neuron-logit map weights $W_L$. As with the mainline model discussed in the main body and discussed in Appendix C.2.1, $W_L$ is consistently sparse, providing is evidence that all four are operating in a Fourier basis.

| Seed | Test Loss | Loss (Key frequencies removed) | Loss (All other frequencies removed) |
|------|-----------|-------------------------------|--------------------------------------|
| 1 | $2.07 \cdot 10^{-7}$ | $6.5 \cdot 10^{0}$ | $5.7 \cdot 10^{-8}$ |
| 2 | $2.1 \cdot 10^{-7}$ | $1.1 \cdot 10^{1}$ | $6.2 \cdot 10^{-8}$ |
| 3 | $2.05 \cdot 10^{-7}$ | $6.7 \cdot 10^{0}$ | $5.5 \cdot 10^{-8}$ |
| 4 | $2.33 \cdot 10^{-7}$ | $6.8 \cdot 10^{0}$ | $6.0 \cdot 10^{-8}$ |

Table 4: As discussed in Appendix C.2.1, ablating the key frequencies for each of the networks reduces performance to worse than chance, while ablating all other frequencies *improves* performance.

**Confirming that the other seeds use the Fourier Multiplication Algorithm.** In Figure 16, we show the norms of the Fourier components of the embedding matrix $W_E$ for each of the 4 other random seeds. As with the mainline model, the matrices are sparse in the Fourier basis. In Figure 17, we show the norms of the Fourier components of the neuron-logit map $W_L$ for the 4 other random seeds. The matrices are sparse in the Fourier basis, enabling us to identify 3 or 4 key frequencies for each of the seeds. Again, note that the specific frequencies differ by seed.

Using the key frequencies identified in the neuron-logit map, we repeat the experiment in Section 4.2, where we "read off" the MLP activations in the 6 or 8 directions corresponding to the key frequencies. As with our mainline model, this lets us identify the trigonometric identities for $\cos(w_k(a + b))$ and $\sin(w_k(a + b))$ being computed at the MLP layer. We confirm that the trigonometric identities are a good approximation by approximating the activations with a single term of the form $\cos(w_k(a + b))$ or $\sin(w_k(a + b))$—as with the mainline model, the fraction of variance explained is consistently close to 100%.

Next, we ablate the key frequencies from the logits as in Section 4.4 and report the results in Table 4. As with the mainline model, ablating all of the key frequencies reduces performance to worse than chance, while ablating everything *but* the key frequencies *improves* test performance.

**Progress measures and grokking.** Finally, we confirm the progress measure and grokking results from the mainline model on other runs with the same architecture. In Figure 18, we display the train, test, and restricted loss for each of the four other random seeds. In Figure 19, we display the Gini coefficients of the Fourier components of the embedding matrix $W_E$ and the neuron-logit map $W_L$ for each of the four other random seeds. The shape of the curves are very similar to those of the mainline model, allowing us to divide grokking on these models into the same three phases

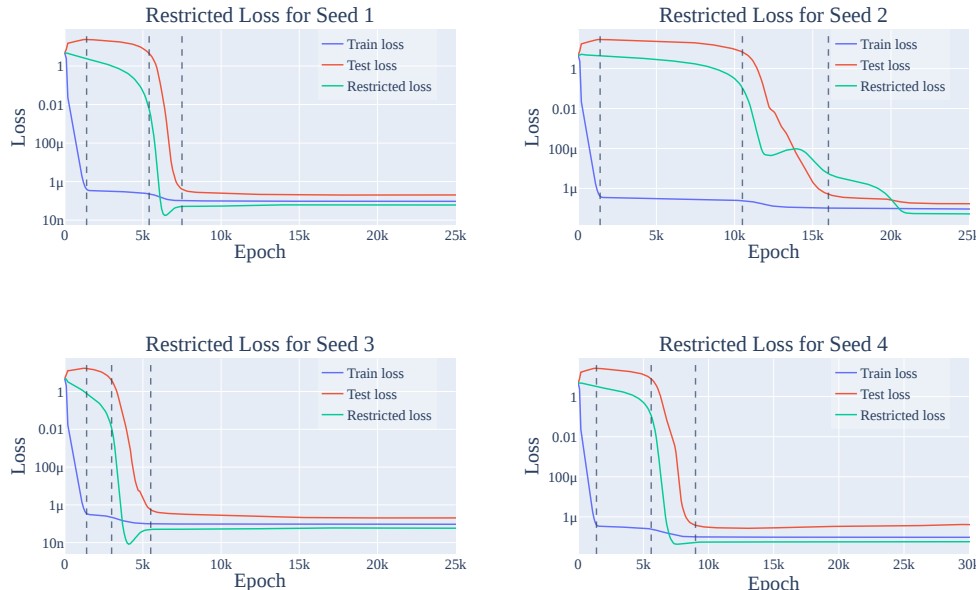

Figure 18: The train, test, and restricted loss for each of the four other random seeds described in Appendix C.2.1. The lines delineate the 3 phases of training: memorization, circuit formation, and cleanup (and a final stable phase). As with the mainline model, restricted loss consistently declines prior to train loss. Note that while the shapes of the loss curves are similar to each other and those of the mainline model, the exact time that grokking occurs (and thus the dividers between the phases of grokking) differ by random seed. Interestingly, memorization is complete by around 1400 steps for all five runs.

identified in the main text. Interestingly, while all of the models complete memorization by around 1400 epochs, circuit formation and cleanup occur at different times.

### C.2.2 RESULTS FOR OTHER EXPERIMENTAL SETUPS

In this section, we provide further evidence that small transformers grok on the modular addition task, by varying the size of the network, the amount of training data, and the size of the prime $P$.

**1-Layer Transformers with Varying Fractions of Training Data.** We find that grokking occurs for the modular addition task with $P = 113$ for many data fractions (that is, the fraction of the $113 \cdot 113$ pairs of inputs that the model sees during training), as shown in Figure 20. Smaller amount lead to slower grokking, but sufficiently large fractions of data ($\geq 60\%$) lead to immediate generalization, as shown in Figures 20 and 21.

As with the results in Appendix C.2.1, all of the 1-layer transformers in this section also converge to using the Fourier multiplication algorithm.

**2-Layer Transformers.** As shown in Figure 22, 2-layer transformers also exhibit some degree of grokking. However, this is complicated by the slingshot mechanism (Thilak et al., 2022). We display the excluded loss of a 2-layer transformer in Figure 23 and find it shows a similar pattern to the mainline 1-layer transformer, in that it improves relatively smoothly *before* grokking occurs.

**Smaller and larger primes.** We also examined smaller and larger prime moduli. For $P = 53$ (Figure 24), we explored a variety of weight decays to observe grokking in the small prime case. With the original weight decay setting of $\lambda = 1$, we found that the models never generalized. However, increasing the weight decay to $\lambda = 5$ does allow the model to grok. We speculate that

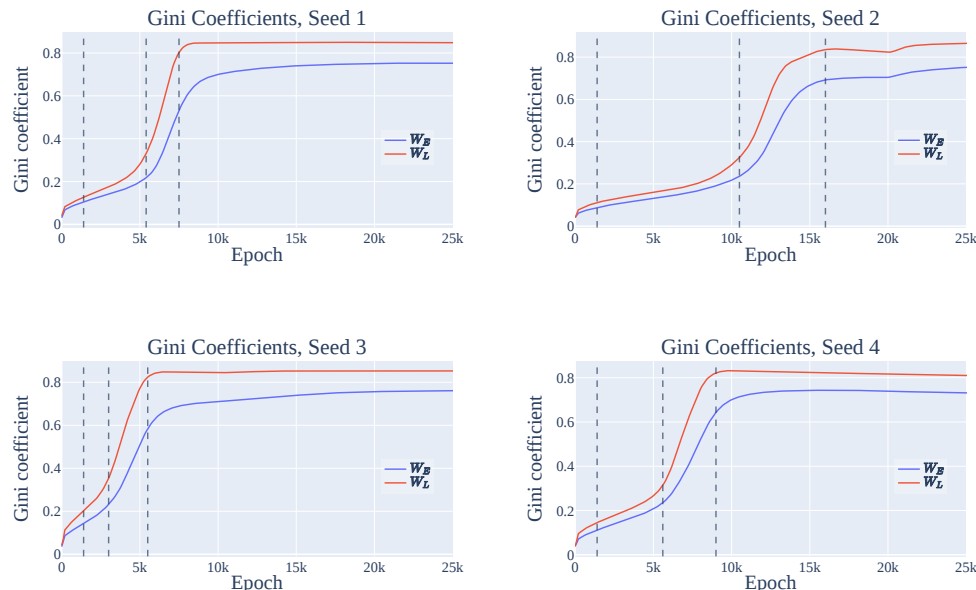

Figure 19: The Gini coefficients (a measure of sparsity) of the Fourier components of the embedding matrix $W_E$ and the neuron-logit map $W_L$ for each of the four other random seeds. The lines delineate the 3 phases of training: memorization, circuit formation, and cleanup (and a final stable phase). As with the mainline model, sparsity increases slowly during memorization and circuit formation, and then quickly during cleanup.

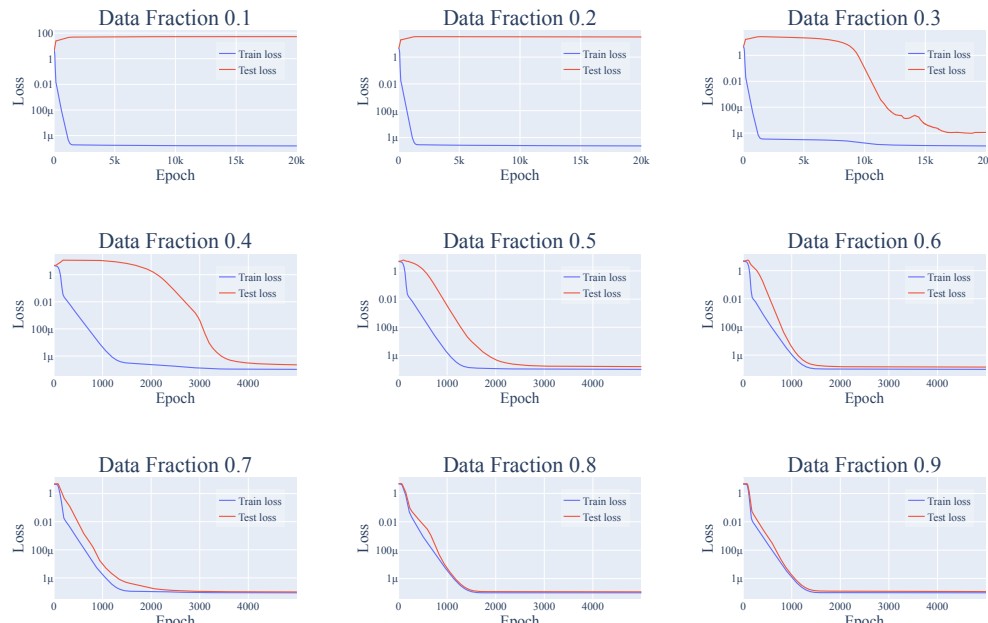

Figure 20: Training and test losses for a 1-layer transformer on the modular addition task with $P = 113$, with varying fractions of the $113 \cdot 113$ pairs of possible inputs used in training. Grokking occurs when between $30 - 50\%$ of the dataset is used during training and lower fractions of data lead to slower grokking. Using $\geq 60\%$ data leads to immediate generalization, while using $10\%$ or $20\%$ of the data doesn't lead to grokking even after 40k epochs. Note the different x-axes: we only show 5k epochs for the runs with data fraction $\geq 40\%$ for more detail.

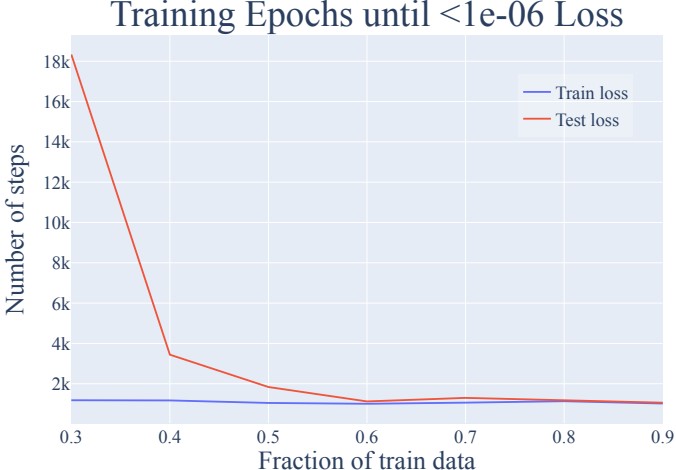

Figure 21: Number of steps for train/test loss to be $< 10^{-6}$, as a function of the amount of training data. While train loss immediately converges to below $10^{-6}$ for all data fractions, generalization takes significantly longer with lower fractions of data. Note that the plots for other thresholds are also qualitatively similar.

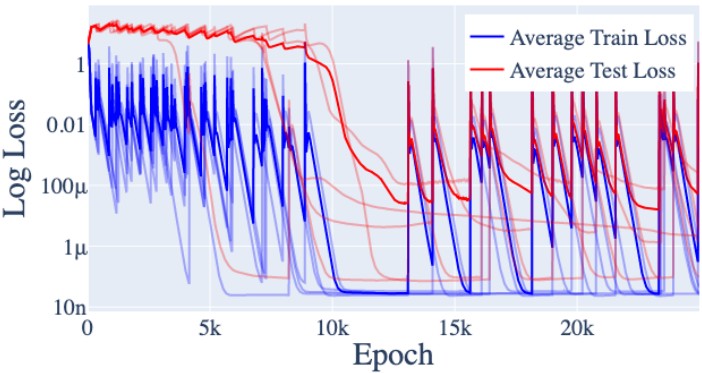

Figure 22: Training and test loss for a 2-layer version of the original architecture. Average across 5 random seeds is in bold.

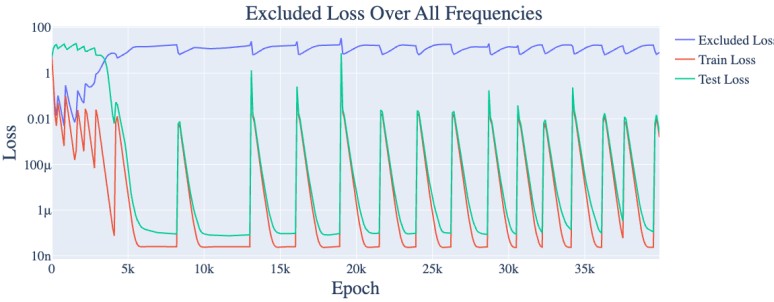

Figure 23: Training, test, and full excluded loss for a 2-layer version of the original architecture. One random seed chosen for readability.

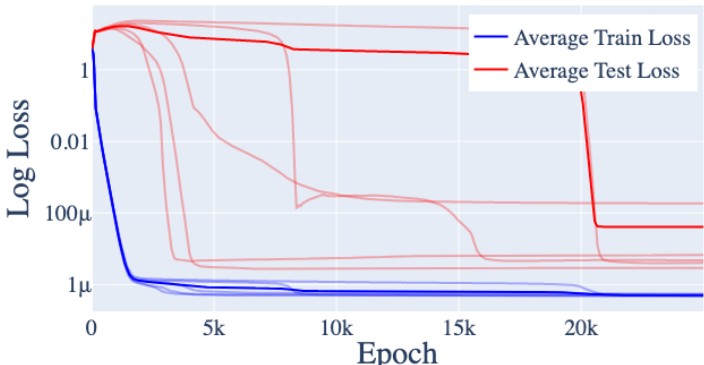

Figure 24: The training and test losses for $P = 53$ and all other hyperparameters except weight decay ($\gamma = 5$) the same as the main training run discussed in the paper. The averages are bold, and all contributing runs are partially transparent. Note that grokking occurs.

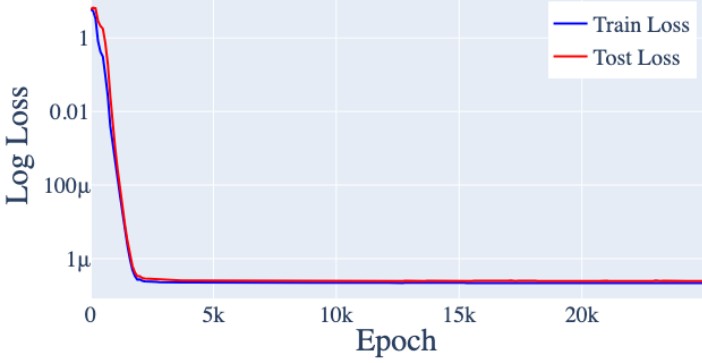

Figure 25: The training and test losses for $P = 401$ and all other hyperparameters the same as the main training run discussed in the paper. Grokking doesn't occur (the model generalizes immediately), even across a variety of weight decays.

this is because the memorization solution is significantly smaller (since there are only $53 \cdot 53$ total pairs), thereby requiring more aggressive weight decay for the generalizing solution to be favored.

For $P = 109$, we saw exactly the same behavior as with the mainline model.

For $P = 401$ (Figure 25), we could not get grokking, even by varying the weight decay parameter $\lambda \in \{0.3, 0.5, 1, 3, 5, 8\}$. Instead, the model immediately learns the generalizing solution. We believe this is because the amount of data seen by the model is greatly increased compared to the $P = 113$ case (from 30% of $113 \cdot 113$ pairs to 30% of $401 \cdot 401$ pairs), thereby favoring the generalizing solution from the start. We then trained 3 models each using 5%, 10%, 20% of the pairs of training data with $\lambda = 1$, and found that the models trained on 5% and 10% of the data immediately overfit and never generalized, while the models trained on 20% of the data also generalized immediately.

### C.2.3 GENERALIZING MODELS CONSISTENTLY USE THE FOURIER MULTIPLICATION ALGORITHM

For each of the models in Appendix C.2.2 that achieve low test loss, we repeated the analysis performed in the mainline model, and summarize the results in Table 5. We list their key frequencies, Gini coefficients, and relevant FVEs. We find that every model trained with weight decay and that generalizes correctly implements some variation of the Fourier multiplication algorithm.

Interestingly, the embedding and unembedding matrices of the models trained with dropout are *not* sparse in the Fourier basis, and the logits for the $p = 0.2$ models are not as well explained by a sum of cosines as the other models (likely because the $p = 0.2$ models are simply worse at the task). We speculate that this is likely due to a combination of insufficient training epochs (as dropout models seem to take much longer to grok) and the inherent need for redundancy for networks trained via dropout.

As with the mainline model, we ignore the final skip connection (around the final MLP), as all of the generalizing models studied do not suffer significant performance penalties if the skip connection is zero or mean ablated (Table 6).

## D ADDITIONAL RESULTS ON GROKKING

### D.1 BOTH REGULARIZATION AND LIMITED DATA ARE NECESSARY FOR GROKKING

As discussed in Section 7 and Appendix C.2, the weight decay and the amount of data seem to have a strong effect on whether grokking occurs. To confirm this, we experiment with removing weight decay and varying the amount of data on 1-layer transformers. In Figure 26, we give the training, test, and full excluded loss for a typical training run with $\lambda = 0$ (no weight decay). As the figure shows, no grokking occurs, and excluded loss does not increase, suggesting that the model does not form the circuit for generalizing algorithm at all.

In Figure 20, we show the test loss curves for models trained with weight decay $\lambda = 1$ and on various fractions of the data. Though all the train losses are approximately the same—that is, they memorize at the same rate, models trained on smaller fractions of data take longer to grok.

In Figure 27, we display the test and train loss of models trained with $\lambda = 0.3$ and $\lambda = 3.0$. Smaller amounts of weight decay lead to slower grokking, while larger amounts of weight decay lead to faster grokking—on average, it takes around 3k epochs for models to grok with weight decay $\lambda = 0.3$, 5-10k epochs for the models to grok with weight decay $\lambda = 1.0$, and 20k epochs for the models to grok with weight decay $\lambda = 3.0$.

Finally, we test whether other forms of regularization can also induce grokking. We replaced weight decay with the following types of regularization while keeping all other hyperpameters the same:

1. **Dropout** We add dropout Srivastava et al. (2014) to the MLP neurons, with $p \in \{0.2, 0.5, 0.8\}$. That is, for each individual neuron, we set it to $0$ with probability $p$ during training, and also multiply the outputs of the other neurons by $\frac{1}{1-p}$.

| Model | Test Loss | Gini($W_E$) | Gini($W_L$) | Key Frequencies | Logit FVE | MLP FVE |
|---|---|---|---|---|---|---|
| 40% Training Data | $1.98 \cdot 10^{-7}$ | 0.76 | 0.79 | [17, 43, 49, 55] | 94.9% | 83.3% [26.1%] |
| 50% Training Data | $1.68 \cdot 10^{-7}$ | 0.75 | 0.77 | [2, 17, 31, 41, 44] | 91.2% | 85.2% [28.2%] |
| 60% Training Data | $1.23 \cdot 10^{-7}$ | 0.79 | 0.84 | [2, 23, 34, 51] | 96.4% | 95.7% [1.4%] |
| 70% Training Data | $9.85 \cdot 10^{-8}$ | 0.80 | 0.91 | [14, 15, 26] | 99.0% | 98.9% [0.4%] |
| 80% Training Data | $5.83 \cdot 10^{-7}$ | 0.62 | 0.80 | [38, 41] | 63.9% | 94.1% [2.5%] |
| 90% Training Data | $1.11 \cdot 10^{-7}$ | 0.79 | 0.88 | [3, 26, 34, 43] | 98.6% | 98.7% [0.3%] |
| 2 Layer Transformer | $9.54 \cdot 10^{-7}$ | 0.59 | 0.80 | [14, 18, 29] | 91.8% | 95.2% [1.9%] |
| 2 Layer Transformer | $4.41 \cdot 10^{-5}$ | 0.55 | 0.73 | [7, 12, 35, 49] | 86.1% | 86.2% [6.4%] |
| 2 Layer Transformer | $6.50 \cdot 10^{-2}$ | 0.66 | 0.80 | [4, 9, 28] | 88.5% | 85.4% [5.9%] |
| 2 Layer Transformer | $4.18 \cdot 10^{-2}$ | 0.56 | 0.76 | [4, 5, 15, 54] | 91.4% | 81.2% [17.8%] |
| 2 Layer Transformer | $1.75 \cdot 10^{-2}$ | 0.68 | 0.71 | [3, 4, 13, 30, 38] | 84.0% | 71.9% [19.5%] |
| $P = 53$ | $3.00 \cdot 10^{-4}$ | 0.61 | 0.68 | [6, 9, 16, 21] | 91.2% | 90.2% [5.8%] |
| $P = 53$ | $1.03 \cdot 10^{-4}$ | 0.56 | 0.72 | [4, 13, 16] | 94.8% | 93.1% [6.4%] |
| $P = 53$ | $1.21 \cdot 10^{-5}$ | 0.66 | 0.79 | [13, 22, 23] | 98.2% | 97.6% [0.9%] |
| $P = 53$ | $3.95 \cdot 10^{-6}$ | 0.66 | 0.74 | [3, 14, 15] | 88.5% | 91.8% [4.6%] |
| $P = 53$ | $5.56 \cdot 10^{-6}$ | 0.67 | 0.80 | [10, 14, 22] | 98.1% | 98.3% [0.6%] |
| $P = 109$ | $2.02 \cdot 10^{-7}$ | 0.76 | 0.83 | [6, 7, 22, 25] | 98.0% | 97.3% [1.9%] |
| $P = 109$ | $2.95 \cdot 10^{-7}$ | 0.69 | 0.82 | [8, 14, 29, 32, 41] | 95.2% | 94.7% [2.3%] |
| $P = 109$ | $1.66 \cdot 10^{-7}$ | 0.78 | 0.86 | [13, 23, 39, 45] | 98.5% | 97.6% [0.9%] |
| $P = 109$ | $2.50 \cdot 10^{-7}$ | 0.68 | 0.82 | [8, 13, 32, 41] | 96.8% | 95.5% [2.3%] |
| $P = 109$ | $2.77 \cdot 10^{-7}$ | 0.76 | 0.85 | [29, 37, 38, 49] | 97.9% | 98.1% [0.8%] |
| Dropout $p = 0.2$ | $2.65 \cdot 10^{-1}$ | 0.19 | 0.46 | [1, 4, 7, 17, 22, 33, 40, 49, 55] | 71.3% | 65.0% [17.5%] |
| Dropout $p = 0.2$ | $4.52 \cdot 10^{-1}$ | 0.19 | 0.46 | [3, 8, 19, 28, 32, 34, 40, 44] | 73.3% | 71.4% [10.7%] |
| Dropout $p = 0.2$ | $2.03 \cdot 10^{-1}$ | 0.20 | 0.45 | [4, 5, 32, 38, 41, 44, 49, 50] | 74.2% | 71.1% [10.6%] |
| Dropout $p = 0.5$ | $< 10^{-8}$ | 0.26 | 0.56 | [1, 4, 26, 46, 47, 55] | 89.4% | 88.9% [3.5%] |
| Dropout $p = 0.5$ | $2.01 \cdot 10^{-2}$ | 0.20 | 0.49 | [16, 21, 35, 47, 53] | 88.4% | 88.4% [3.0%] |
| Dropout $p = 0.5$ | $< 10^{-8}$ | 0.25 | 0.54 | [1, 4, 7, 19, 29, 31, 42] | 86.1% | 85.6% [4.0%] |

Table 5: For each of the models in Appendices C.2.3 and D.1 that generalizes to test data, we report the test loss, the Gini coefficients of the norms of the Fourier components of $W_E$ and $W_L$ (Section 5.1), the key frequencies of the network, and the fraction of variance in logits explained by a weighted sum of $\cos(w_k(a + b - c))$s over the key frequencies (Section 4.2).

In addition, we find the components $u_k, v_k$ of $W_L$ that correspond to cosines and sines of the key frequencies, and then report the average fraction of variance of $u_k^T \text{MLP}(a, b)$ and $v_k^T \text{MLP}(a, b)$ explained by a single term of form $\cos(w_k(a + b))$ or $\sin(w_k(a + b))$ respectively (Section 4.2). Numbers in square brackets represent the standard deviation. For 2 Layer models, we use the final layer MLP activations for $\text{MLP}(a, b)$.

We omit test accuracy because every model on this list except for the dropout $p = 0.2$ models achieves $> 99.95\%$ test accuracy, while the dropout $p = 0.2$ models achieve around $99.6\%$ test accuracy.

| Model Type | Loss | Accuracy | Ablated Loss | Ablated Acuracy |
|---|---|---|---|---|
| Varying Data Fraction | $1.83 \cdot 10^{-7} \, (1.65 \cdot 10^{-7})$ | 100% | $7.74 \cdot 10^{-7} \, (6.74 \cdot 10^{-7})$ | 100% |
| 2 Layer Transformer | $1.97 \cdot 10^{-2} \, (2.41 \cdot 10^{-2})$ | 99.6% | $4.63 \cdot 10^{-2} \, (6.72 \cdot 10^{-2})$ | 98.7% |
| $P = 53$ | $5.96 \cdot 10^{-5} \, (8.91 \cdot 10^{-5})$ | 100% | $1.5 \cdot 10^{-4} \, (2.70 \cdot 10^{-4})$ | 100% |
| $P = 109$ | $1.94 \cdot 10^{-7} \, (3.74 \cdot 10^{-8})$ | 100% | $6.53 \cdot 10^{-7} \, (1.41 \cdot 10^{-7})$ | 100% |
| Dropout $p = 0.2$ | $0.215 \, (0.091)$ | 99.7% | $0.205 \, (0.075)$ | 99.7% |
| Dropout $p = 0.5$ | $4.68 \cdot 10^{-3} \, (8.11 \cdot 10^{-3})$ | 100% | $3.6 \cdot 10^{-3} \, (5.82 \cdot 10^{-3})$ | 100% |

Table 6: We confirm that the skip connection around the final MLP layer is not important for performance by mean ablating the skip connection and computing loss and accuracy over the entire dataset for each problem, averaged over all runs. (We report the standard deviation of loss over the runs in parentheses.) While loss does increase a small amount, accuracy remains consistently high and the loss of the ablated model remains low. Results with zero ablations are also similar.

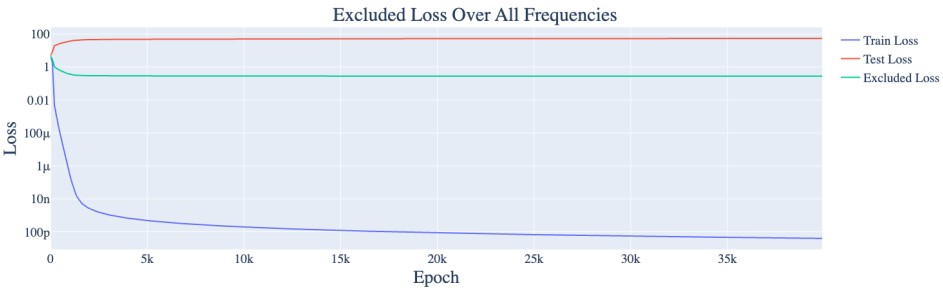

Figure 26: Training, test, and full excluded loss for a 1-layer version of the original architecture without weight decay. One random seed chosen for readability. Note that not having weight decay prevents grokking.

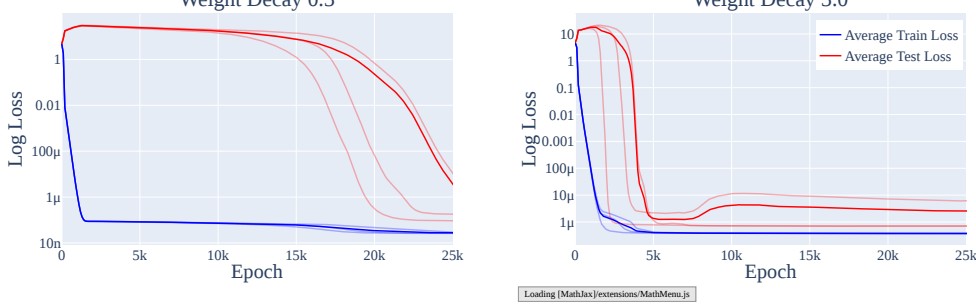

Figure 27: The train and test loss over the course of training with weight decay $\lambda = 0.3$ (left) and $\lambda = 3.0$ (right). Less aggressive weight decay leads to slower grokking.

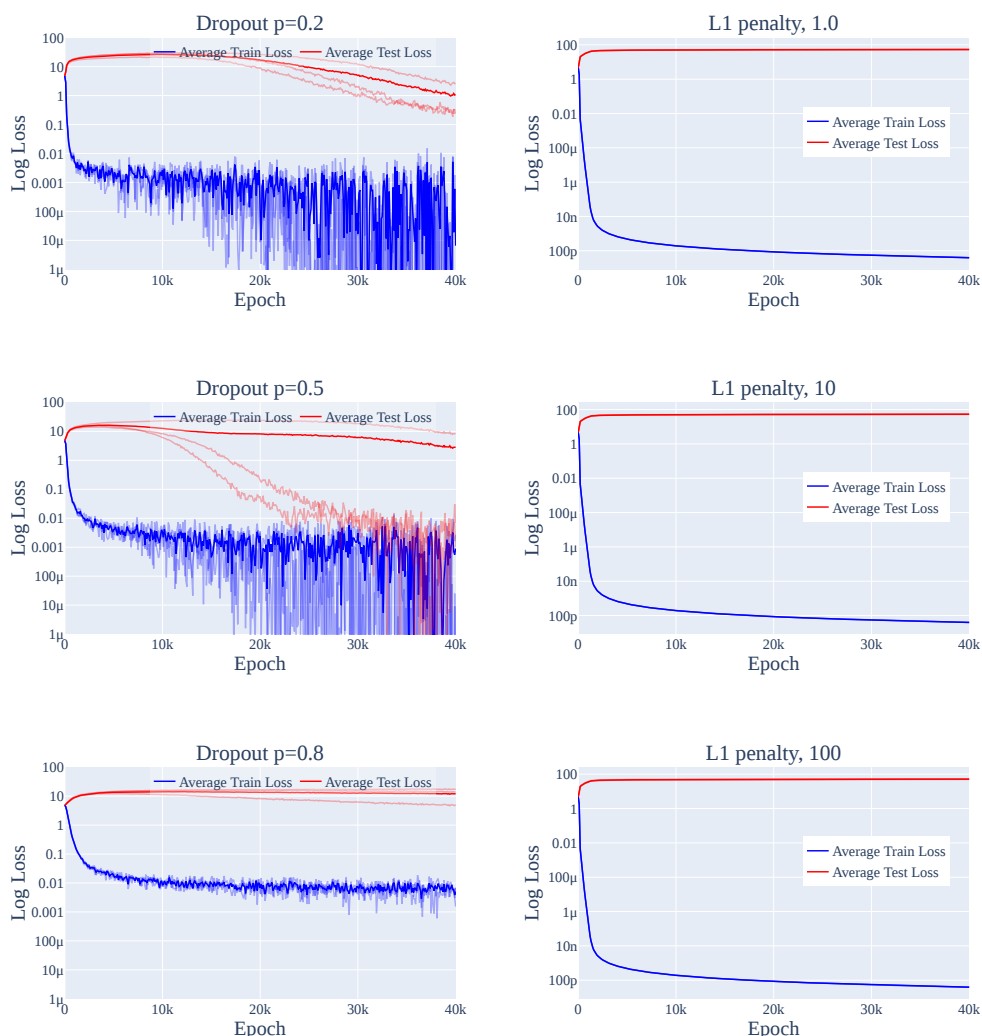

Figure 28: The train and test loss over the course of training with two types of regularization, dropout and $\ell_1$ regularization. Grokking occurs with some runs for dropout but never for $\ell_1$ regularization.

2. $\ell_1$ **Regularization** We add an $\ell_1$ penalty to the loss term. We use $\lambda \in \{1, 10, 100\}$. Note that we *do not decouple* the updates with respect to the $\ell_1$ penalty from optimization steps done with respect to the log loss (as is done for $\ell_2$ regularization via AdamW Loshchilov & Hutter (2017)).

In each case, we ran three random seeds. We show the results in Figure 28. While grokking did not occur with $\ell_1$ regularization, we found that it does occur for all three seeds using dropout with $p = 0.2$ or $p = 0.5$. We speculate that this is because both dropout and weight decay encourage the network to spread out computation (which is required for the Fourier multiplication algorithm), while $\ell_1$ regularization encourages the network to become more sparse in the neuron basis and thus *less* sparse in the Fourier basis, preventing the network from learning the Fourier Multiplication Algorithm.

D.2    THE SLINGSHOT MECHANISM OFTEN OCCURS, BUT IS UNNECESSARY FOR GROKKING

As noted in Section C.2, our 2-layer transformers exhibit significant slingshots (Thilak et al., 2022) during training. We speculate that this is due to how gradients of different scale interact with adaptive

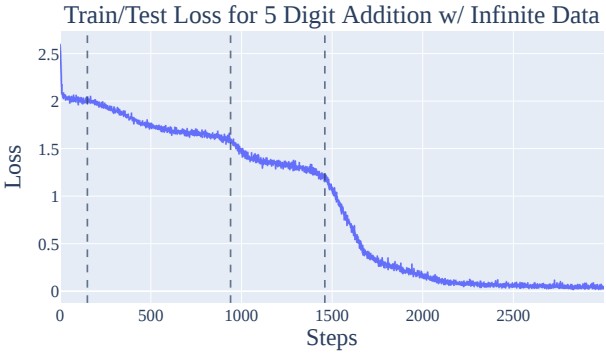

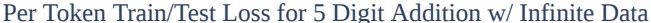

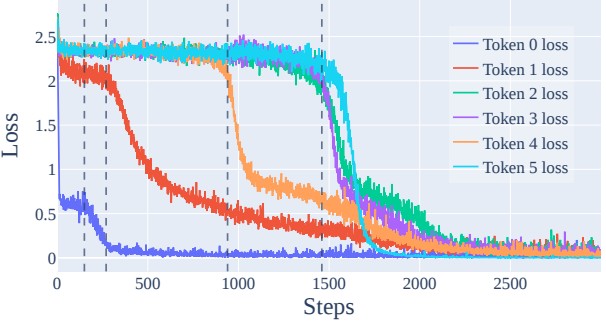

Figure 29: (*Top*) The training/test loss for 5 Digit Addition trained on randomly generated data. Note that training and test loss coincide, as the model does not see repeated pairs.(*Bottom*) The train/test loss *per token* for 5 Digit Addition, trained with randomly generated data at each step. Note that phase changes in the average loss correspond to phase changes in individual tokens, though one phase change (token 1, around step 270) is not visible on the averaged loss as it overlaps with the end of the first phase change (token 0, starting around step 150).

optimizers. We were even able to induce slingshots on a 1-layer by reducing the precision of the loss calculations (as this causes many gradients to round to 0 and thus greatly increases the differences in scale of gradients).

However, as many of our 1-layer models do not exhibit slingshots but nonetheless grok, the slingshot mechanism is unnecessary for grokking to occur, in the presence of weight decay or other regularization. We speculate that the slingshots of Thilak et al. (2022) (which co-occur with grokking for training runs *without* weight decay) serve as an implicit regularization mechanism that favors the simpler, generalizing solution over the more complicated

## D.3 ADDITIONAL EVIDENCE FROM OTHER ALGORITHMIC TASKS

We now provide addition analysis of grokking phenomena on 3 additional algorithmic tasks and confirm that limited data is an important part of grokking:

1. **5 digit addition.** We sample pairs of random 5 digit numbers and have the model predict their sum
2. **Predicting repeated subsequences.** We take a uniform random sequence of tokens, randomly choose a subsequence to repeat, and train the model to predict the repeated tokens.
3. **Skip trigram.** We feed in a sequence of tokens from 0 to 19, of which exactly one is greater than or equal to 10, and the model needs to output the token that is $\geq 10$. This can be solved with learning 10 skip trigrams.

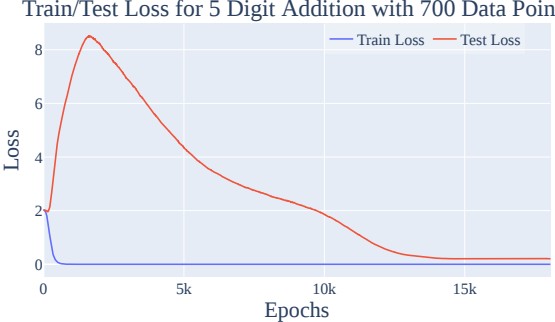

Figure 30: The train and test loss for 5 Digit Addition trained on 700 data points. Unlike the infinite, randomly generated data case, this shows both a sharp phase change and clear train test divergence.

We use a 1-layer full transformer for 5-digit addition, a 2-layer attention only transformer for predicting repeated subsequences, and a 1-layer attention only transformer for the skip trigram task. Otherwise, we use the same hyperparameters as in the mainline model.

**5 Digit Addition** We first consider the case where we train on the approximately infinite data regime. For each minibatch, we randomly new sample 5 digit numbers. We report the results in Figure 29. Train loss coincides with test loss, so grokking does not occur, as the model almost never sees the same pair of 5 digit numbers twice, with $10^{10}$ such pairs. Interestingly, the various small bumps in Figure 29 correspond to the model learning how to calculate each of the 6 tokens in the output. However, grokking does occur when we restrict the model to only see 700 data points, as shown in Figure 30.

**Repeated subsequence** As with the 5-digit addition task, we find that restricting the amount of data is necessary and sufficient for grokking on the repeated subsequence task. In Figure 31, the model sees new data at every step exhibits no grokking. In contrast, clear grokking occurs when we restrict the model to only see 512 data points in Figure 32.

**Skip trigram** As with the previous tasks, we find that restricting the amount of data is necessary and sufficient for grokking on the skip trigram task. The model that sees new data at every step exhibits no grokking in Figure 33. Meanwhile, the model restricted to only see 512 data points exhibits clear grokking in Figure 34.

Taken together, these results echo the importance of limited data for grokking.

## E  FURTHER SPECULATIONS ON GROKKING

### E.1  AN INTUITIVE EXPLANATION OF GROKKING

In this section, we speculate on what might be happening "under the hood" when a model groks and explore why this phenomena happens. The evidence is only suggestive, so this a promising direction for future research.

Grokking occurs when models, trained on algorithmic tasks with certain hyperparameters, initially overfit the training data where train loss significantly improves while test loss worsens and the two diverge. But later in training, there is a sudden improvement in test loss, so test and train loss converge. In contrast to Power et al. (2022) but in line with Liu et al. (2022), grokking does *not* occur when both train and test loss improve together without the initial divergence, as shown in many of the figures in this paper, for example Figures 2 and 18.

The core issue is that the model has two possible solutions: memorization (with low train loss and high test loss) and a generalization (with low train loss *and* low test loss). In our case, the

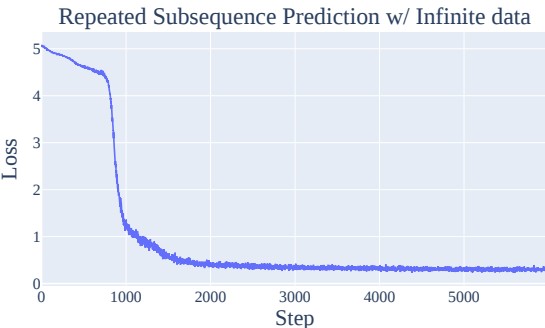

Figure 31: The training/test loss for repeated subsequences trained on randomly generated data. Note that training and test loss coincide, as the model does not see repeated pairs. There sharp phase change corresponds to the model forming induction heads. (Olsson et al., 2022)

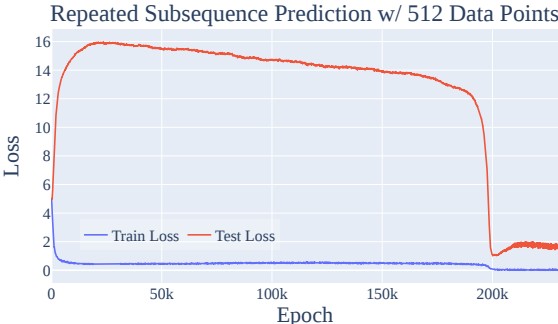

Figure 32: The train and test loss for the repeated subsequence task, trained on 512 data points. Unlike the infinite, randomly generated data case, this shows both a sharp phase change and clear train test divergence.

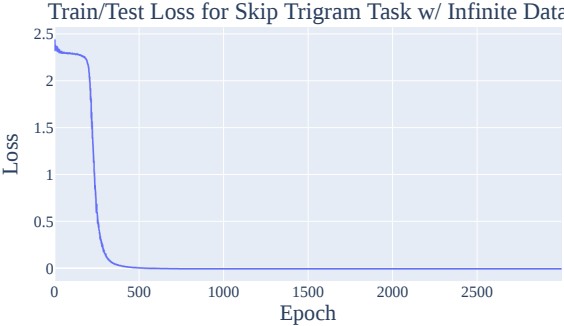

Figure 33: The training/test loss for the skip trigram task, trained on randomly generated data. Note that training and test loss coincide, as the model does not see repeated pairs. The sharp phase change corresponds to the network learning all of the skip trigrams.

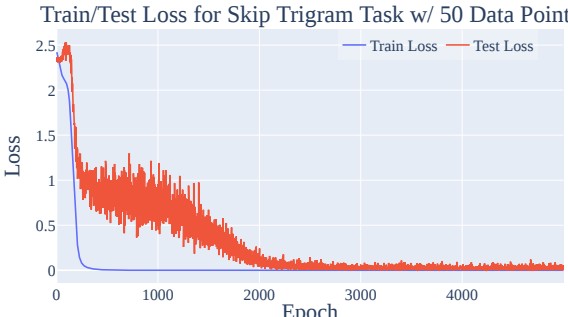

Figure 34: The train and test loss for the skip trigram task, trained on 512 data points. Unlike the infinite, randomly generated data case, this shows both a sharp phase change and clear train test divergence.

Fourier Multiplication Algorithm is the generalization solution. Intuitively, with very little training data, the model will overfit and memorize. With more training data, the model must generalize or suffer poor performance on both train and test loss. Since neural networks have an inductive bias favoring "simpler" solutions, memorization complexity scales with the size of the training set, whereas generalization complexity is constant. The two must cross at some point! Yet, the surprising aspect of grokking is the abrupt shift during training, when the model *switches* from memorization to generalization.

The other component of grokking is phase transitions - the phenomena where models trained on a certain task develop a specific capability fairly rapidly during a brief period of training, as shown for the case of induction heads forming in transformer language models in Olsson et al. (2022) and our results in Appendix D.3. That is, rather than slowly forming that capability over training, the model rapidly goes from being bad at it to being good at it. One interpretation of a phase transition is that there's some feature of the loss landscape that makes the generalising solution harder to reach - rather than a smooth gradient for the model to follow, it instead initially finds it difficult to make progress, but then crosses some threshold where it can rapidly make progress.

Therefore, grokking occurs with phase transitions, limited data, and regularization. Models exhibit phase transitions despite having enough training data to avoid overfitting. Regularization (weight decay in our case) favors simpler solutions over complex ones. The model has enough data to marginally prefer generalization over memorization. The phase transition indicates that generalization is "hard to reach" while the model has no problems with memorization. But as it memorizes, the network becomes more complex until the weight decay prevents further memorization then moves towards equilibrium. The gradient to memorize balances the gradient towards smaller weights. With generalization, the model is incentivized to both memorize and simplify. Strikingly, it is capable of both while maintaining a somewhat constant training performance in this circuit formation phase. Next, as the model approaches generalization, the memorization weights are removed in the cleanup phase. The cost from complexity outweighs the benefit from lower loss. Due to the phase transition during this training period, as model's progress towards generalization accelerates, the cleanup rate sharpens as well.

A model that learns a perfect solution and is trained with weight decay has competing incentives: larger weights (for more extreme logits and thus lower loss) and smaller weights (from weight decay). So for any solution and any level of weight decay, there will always be a level of train loss where these two forces equilibrate. Thus, memorization is not necessarily a "simpler" solution than generalization. The key is that generalization will have smaller weights *holding train loss fixed*. In fact, weight decay should be expected to equilibrate at a slightly lower train loss in generalization, since the base solution is simpler. This matches what we observe in practice. [4]

---

[4]One subtlety: the grokking phenomena is often incorrectly summarized as "the model learned to generalize even after achieving zero loss." Zero loss does not exist with cross-entropy loss. Although the model achieves

### E.2 HYPOTHESIS: PHASE TRANSITIONS ARE INHERENT TO COMPOSITION

A promising line of work in the growing field of mechanistic interpretability suggests that models form *circuits* (Cammarata et al., 2020) – clean interpretable algorithms formed by subnetworks of the model, such as curve detectors (Cammarata et al., 2020) in image classification networks and induction heads (Elhage et al., 2021; Olsson et al., 2022) in LLMs. This is surprisingly true! A circuit represents the model learning an algorithm, a fundamentally discrete thing; each step in the algorithm only makes sense if the other steps are present. But neural networks are fundamentally continuous, trained to follow gradients towards lower loss and struggle to jump to new optima without following a smooth gradient. So how can a model learn a discrete algorithm?

As a concrete example, let's consider the case of induction heads in LLMs. There is a subnetwork of a next-token prediction autoregressive language model that learns to continue repeated subsequences. It detects whether the current token occurred earlier in the context. If so, it predicts the same token after that previous occurrence will also come next. The circuit consists of a previous token head, which attends to each previous token and copies the context of the previous token to the current token, and an induction head which attends to the token *after* a previous occurrence of the current token. The induction head composes with the previous token head by forming a query vector representing the current token and a key vector representing the previous token head's output using K-Composition, the context of the previous token. It attends to a token where this query and key match.

This circuit significantly improves loss but only in the context of the other heads present. Before either head is present, no gradient encourages the formation of either head. At initialization, we have neither head, so gradient descent should never discover this circuit. Naively, we might predict that neural networks will only produce circuits analogous to linear regression, where each weight will marginally improve performance as it continuously trains. And yet in practice, neural networks indeed form such sophisticated circuits, involving several parts interacting in non-trivial, algorithmic ways. So how can this be?

A few possible explanations:

- **Lottery tickets (Frankle & Carbin, 2018):** Initially, each layer of the network is the superposition of many partial circuit components, and the output of each layer is the average of the output of each component. The full output of the network is the average of many different circuits, with significant interference from non-linear interaction. Some of these circuits are systematically useful to reducing loss, but most aren't. Gradients for useless circuits will have zero mean, while gradients for useful circuits will have non-zero mean, with a lot of noise. SGD reinforces relevant circuits and suppresses useless ones, so circuits will gradually form.

- **Random walk:** The network wanders randomly around the loss landscape until it encounters a half-formed previous token head and induction head that somewhat compose. This half-formed circuit becomes useful for reducing loss, so gradient descent completes the circuit.

- **Evolution:** A similar mystery arises from how organisms develop sophisticated machinery, like the human eye. Each part is only useful in the context of other parts. A compelling explanation is a component first developed that was somewhat useful in its own right, like a light-detecting membrane. It was reinforced as a useful component. Then, later components developed depending on the first, like the lens of the eye.

Evolution is a natural explanation, However, based on our toy tasks, it cannot be the whole story. In the repeated subsequence task, we have a sequence of uniform randomly generated tokens, apart from a repeated subsequence at an arbitrary location, e.g. 7 2 8 3 1 9 3 8 3 1 9 9 2 5 END. This means all pairs of tokens are independent, apart from pairs of equal tokens in the repeated subsequence. In particular, this means that a previous token head can never reduce loss for the current token. The previous token will always be independent of the next token. So a previous token head is only useful in the context of an induction-like head that completes the circuit. Likewise, an induction head relies

---

perfect *accuracy*, it is trained to optimize loss not accuracy. This means the model is *always* incentivized to further improve. In particular, the easiest way to improve performance with perfect accuracy is by scaling up the logits. This lowers the temperature and pushes the softmax closer to an argmax.

on K-composition with a previous token head and so cannot be useful on its own. Yet the model eventually forms an induction circuit!

A priori, the random walk seems insufficient on its own. An induction circuit is relatively complicated, representing a small region in model space. So a random walk is unlikely to stumble upon it. Concretely, in our modular addition case, progress measures show significant hidden progress pre-grokking, indicating the model did not stumble upon the solution by chance.

Thus, the lottery ticket hypothesis seems the most explanatory. An induction head is useless without a previous token head but might be slightly useful when composing with a head that uniformly attends to prior tokens, since part of its output will include the previous token! Nevertheless, we suspect that all explanations contribute to the entire picture. This seems most plausible if the uniform head just so happens to attend a bit more to the previous token via a random walk.

Returning to phase transitions, the lottery ticket-style explanation suggests that we might expect phase transitions as circuits form. Early in circuit formation, each part of the circuit is rough, so the effect on the loss of improving any individual component is weak, meaning gradients will be small. As each component develops, other components will become more useful, meaning all gradients will increase together non-linearly. As the circuit nears completion, we should expect an acceleration in the loss curve for this circuit, resulting in a phase transition.

## F  FURTHER DISCUSSION ON USING MECHANISTIC INTERPRETABILITY AND PROGRESS MEASURES FOR STUDYING EMERGENT PHENOMENA

While we find approach of using mechanistic interpretability to define progress measures relatively promising, there remains significant uncertainty as to how scalable existing mechanistic interpretability approaches really are. Broadly speaking, depending on the success of future mechanistic interpretability work, we think there are three methods through which mechanistic interpretability and progress measures can help with understanding and predicting emergent phenomena:

1. If mechanistic interpretability can be scaled to large models to the level where we can understand the mechanisms behind significant portions of their behavior, we could perform the same style of analysis as was done in this work. We believe it's currently unclear as to whether or not mechanistic interpretability will successfully scale to large models to this extent (or even if there exist human-understandable explanations for all of their sophisticated behavior). That being said, in cases where mechanistic interpretability does recover human-understandable mechanisms, we could simply use the parts of the mechanism as progress measures.

2. If future mechanistic interpretability can only recover parts of the mechanism of larger models (as in Wang et al. (2022)) and can only generate comprehensive understanding of the mechanisms of smaller models, we might still be able to use our understanding from smaller models to guide the development measures that track parts of the behavior of the larger model. We find this scenario relatively plausible, as existing mechanistic interpretability work already allows us to recover fragments of large model behavior and understand these fragments by analogy to smaller models. For example, Olsson et al. (2022) use this approach to understand the emergence of in-context learning in medium-sized language transformers.

3. Even if mechanistic interpretability fails to recover understandable mechanisms at all on large models, we might still be able to derive progress measures that don't require human understanding. For example, if we end up with automated mechanistic interpretability (that nonetheless still fails to recover human-understandable mechanisms), we might be able to use the outputs of those opaque processes.

   Another approach is task-independent progress measures: if we can discover progress measures that don't depend on the task, perhaps using many small, interpretable models as testbeds, we might be able to apply these progress measures to large models.

That being said, we think the future work outlined in Section 6 is necessary to successfully apply our approach to predict and understand emergent behavior in existing large language models, and so remain cautiously optimistic.

