# OpenReview forum: "Progress measures for grokking via mechanistic interpretability"
_ICLR.cc/2023/Conference — ICLR 2023 notable top 25%_

### Official Review · Reviewer_WFHw · 2022-10-23

**Confidence:** 4
**Correctness:** 3
**Technical Novelty And Significance:** 3
**Empirical Novelty And Significance:** Not applicable
**Recommendation:** 8

**Clarity, Quality, Novelty And Reproducibility:**

- The paper is clearly written and of reasonably high technical quality.
- The exact approach, including the actual progress metrics and how they are interpreted and related to different learning phases, appears original, though the methodology is very similar to earlier works.
- Good reproducibility: A Colab notebook with code is provided.

**Strength And Weaknesses:**

Strengths:
- The paper studies a critical problem of interpreting transformer networks.
- The presentation is well-written and logical.
- The method is intuitive and yields non-trivial results that may provide tools for interpreting larger networks.

Weaknesses:
- My major concern is the generalizability of the approach to more complex networks and training data. There is a clear risk that these clean results cannot be applied outside toy problems like the one presented here.

**Summary Of The Paper:**

The paper presents a technique for analysing transformer networks by reverse engineering the behaviors of a simple network into their constituent components, similarly to the mechanistic interpretability employed in e.g. (Elhage et al. 2021). The technique is applied on a network trained with known trigonometric functions that lends themselves for relatively transparent analysis. The paper purportedly shows distinct memorization and 'grokking' phases of learning happening in the target network. Finally, with thus framed progress in learning, 'progress' metrics are defined and measured such that they align with that progress.

**Summary Of The Review:**

While I am genuinely concerned about generalizability of the results with this simple training setup, I would approve this as a case where one has to work upwards from a simple initial model. I consider it sufficient that the paper suggests a specific methodology and illustrates it with the given limited example. I would not claim the authors actually prove that those properties and training dynamics are applicable across a wider range of networks and training scenarios, but anyway that is not necessary in this case.

Hence, I find this an interesting paper that may contribute at least to the transformer interpretation methodologies, and possibly to other networks as well.

---

> ### Author Response · Authors · 2022-11-17
> **Response to reviewer WFHw**
>
> Thanks for the review! We’re happy to see that you recognize the potential in this approach, and feel that our presentation is well-written and logical.
>
>
> > The technique is applied on a network trained with known trigonometric functions that lends themselves for relatively transparent analysis.
>
> We did not know the trigonometric functions ahead of time; we trained the network on modular addition and recovered the Fourier Multiplication by examining the weights of the network (specifically, we noticed the periodicity in the weights and activations after visualizing them). We do agree that the idea of using trigonometric identities (or more generally, using the roots of unity) to perform modular addition is very well known, though.
>
> > My major concern is the generalizability of the approach to more complex networks and training data. There is a clear risk that these clean results cannot be applied outside toy problems like the one presented here.
>
> We agree to some extent: as we mention in the discussion, we think there are several clear obstacles toward using mechanistic interpretability to derive progress measures for models in practice. Notably, we mention “larger models and realistic tasks” as an area of future work.
>
> Generally speaking, we think there are three approaches for extending this work to using progress measures to understand emergent phenomena:
> 1. If mechanistic interpretability can be scaled to large models to the level where we can understand the mechanisms behind significant portions of their behavior, we could perform the same style of analysis as was done in this work, and use the parts of the mechanism as a progress measure.
> 2. If future mechanistic interpretability can only recover parts of the mechanism of larger models and can only generate comprehensive understanding of the mechanisms of smaller models, we might still be able to use our understanding from smaller models to guide the development measures that track parts of the behavior of the larger model. For example, Olsson et al 2022 do this to understand the emergence of in-context learning in medium-sized language transformers.
> 3. Even if mechanistic interpretability fails to recover understandable mechanisms at all on large models, we might still be able to derive progress measures that don’t require human understanding. For example, if we end up with automated mechanistic interpretability (that nonetheless still fails to recover human-understandable mechanisms), we might be able to use the outputs of those opaque processes. Another approach is task-independent progress measures: for example, we’ve attempted some preliminary experiments attempting to derive progress measures by looking at the principle components of variation of the network weights and logits over time*; if approaches like these worked, we might be able to derive task-independent progress measures. The lottery ticket style of approach suggested by Reviewer u853  might also allow us to derive task-independent progress measures.
>
> In addition to the Induction Heads paper (Olsson et al 2022) mentioned above, we think that other recent work in mechanistic interpretability suggests that this approach might scale to less toy tasks. For example, in another concurrent ICLR submission (https://openreview.net/forum?id=NpsVSN6o4ul), the authors were able to discover and interpret a large circuit in GPT-2-small, while Cammarata et al were able to reverse engineer curve detectors in inception (https://distill.pub/2020/circuits/curve-circuits/). So we are cautiously optimistic about this approach.
>
> We’ve added more discussion of this to the appendix of the paper.

---

> > ### Comment · Reviewer_WFHw · 2022-11-27
> > **Reply from reviewer**
> >
> > Thank you for your the reply and the clarification about the role of trigonometric functions in the argument. I find the references you provide also interesting.
> >
> > I remain cautious about the generalizability of the interpretability aspect itself, but generally agree with your 3-fold argument for extensions around progress measures. I have increased my score to more strongly favor acceptance.

---

### Official Review · Reviewer_u853 · 2022-10-23

**Confidence:** 4
**Correctness:** 4
**Technical Novelty And Significance:** 3
**Empirical Novelty And Significance:** 4
**Recommendation:** 8

**Clarity, Quality, Novelty And Reproducibility:**

The paper is well written, and Fig 1 and the summary/outlook helps navigate the complexity of the analysis (though a certain amount of complexity is unavoidable). The quality of the analysis is very high; ablations and control experiments have been performed; most impressively the replacement of parts of the neural net with its exact algorithmic counterpart. Results are novel and original, and have not yet been published in a peer reviewed article as far as I know. In terms of reproducibility, the paper refers to a code release, which is great - it would still be good to include sufficient detail to repeat the experiments in the paper (appendix).

**Strength And Weaknesses:**

**Main contributions, Impact**
1) Identification and analysis of the algorithm that the grokking transformer uses to achieve strong generalization beyond the training set. The main finding is that the network learns to implement modular addition via projections onto and rotations on a circle (Fourier transformations and trigonometric identities). The analysis is sophisticated, intricate, and very well executed - including ablations and controls to verify many steps of the analysis. Impact: while understanding the particular algorithm is interesting (and certainly labor-intensive), I think the even more important finding is to have another rare case-study of deciphering the inner workings of a neural network that generalizes well, to reveal an algorithmically elegant and sensible solution; the work shows that “opening the black box” is indeed possible. Because the case study is so well executed I would rate its potential impact as a widely-cited and pioneering success case high.

2) Design of two metrics, that allow to diagnose and track grokking smoothly over training. The introduction somewhat suggests that such “progress measures” can potentially be used to predict spontaneously emerging capabilities in AI systems in advance. While the paper demonstrates a concrete instance of the existence of such measures, I would almost interpret the current findings as somewhat of a negative result for the goal of developing general measures for predicting emergent capabilities: the progress measures could only be developed in hindsight (after a network with such capabilities already exists) and are highly task-specific. Impact: medium to low

3) Abstract characterization of the grokking phenomenon (via the introduced progress measures), leading to the identification of three separate phases (memorization, circuit formation, cleanup). To me personally the most impactful contribution of the paper - it sheds some light and raises interesting hypotheses on the grokking phenomenon in general, beyond the individual task(s) studied. While it still remains somewhat unclear why the grokking solution cannot develop without the memorization phase, I think the paper characterizes some important parts of that question (onset of circuit formation after memorization, smooth development of circuit while memorizing part is still needed, finally fairly rapid switch over to circuit along with removal of now superfluous memorized information). Impact: medium to potentially very high - if we (as a community) could figure out how to obtain the grokking solution fast and reliably and without going through the memorization phase for a broad range of tasks, then we might see a generational leap in neural network training, potentially enabling strong and robust generalization capabilities that seem out of reach currently. Time will tell whether this work contributes important pieces to that puzzle, but I personally think it raises some very interesting observations, and follow-up questions.

**Strengths**
 *  Sophisticated and intricate mechanistic interpretability case-study. Would itself be sufficient for a publication for me.
 *  Investigation of grokking - confirming and rejecting some previously proposed hypotheses, and raining interesting new ones.
 * Most importantly: confirming the intuition that grokking consists of developing a memorizing solution rapidly, while a slower process builds a strongly generalizing solution in a much smaller sub-network in parallel, and finally a fairly rapid switch away from the memorizing solution once the circuit is fully developed.

**Weaknesses**
 * The paper is limited to being a (well executed) case-study around grokking and mechanistic analysis for a particular task (three more tasks in the appendix and various ablations). This makes very general statements about grokking hard to make; and allows the paper to mainly raise, rather than confirm, hypotheses about grokking in general. Having said that, I think the paper “packs” more than enough to warrant a publication.
 * The current discussion could be a bit expanded and perhaps also be a bit more critical - the intro sounds a somewhat like the current paper is a big leap towards general/task-independent progress measures that allow predicting the onset of arbitrary capabilities (yet the current paper highlights the challenges which make such general measures unlikely to exist).
 * Generally very well written, but a few typos, and the appendix needs perhaps another quick pass.

**Improvements**
1) Does training (with different random seeds) always identify the same key frequencies? If yes, is there a hypothesis why that is the case; if no, please mention it and comment on this being a limitation for the design of more general progress measures.

2) No need to add much, but it would be great to see a slightly expanded critical discussion (particularly the last two points). What does the current paper contribute to the development of (general) progress measures, and how hard/easy are the current obstacles?

3) Suggestion (but given the time-frame I do not expect to see these results): identify the “winning lottery ticket” before and after grokking (i.e. remove any unnecessary weights from the network). (I) What is the size of these tickets (and how does it change during circuit formation and cleanup) - related to Fig. 6 bottom right? (II) If the winning ticket after grokking is re-initialized with its original initialization values (and all other parameters are kept at zero during training), can the network recover the grokking solution (without going through a memorization phase; perhaps even faster than in the original training run; in other words: is it enough to identify the correct sub-circuit structure for the grokking solution or do we need the memorization-part of the network during training such that the grokking solution can develop)?

**Minor comments**

A) The paper talks about the emergence of “capabilities”: maybe the phrasing can be made more precise. If a model can memorize the training data; in some sense it has already acquired (partial) “capabilities”. Maybe better to say that what is meant is capabilities that *generalize* beyond the training data (i.e. the network learns a *general* algorithm to implement the capabilities rather than a partial one that relies mainly on memorization).

B) Typos:

P1, typo: “that detect earlier repeated subsequences.”

P2, punctuation: “toward the generalizing algorithm Liu et al.”

P3, Section 3: Footnote 2 is used twice (once erroneously it seems)

P5, typo: “should contain be rank”

P5, missing space: “Table 3:projecting”

P6, typo: “yet we can the logits”

P8, typo (since -> sense): “it makes since to check”

Fig 6: use same color for Test and Train loss in both top panels

Fig 13, typo (90% twice): “While the majority of the runs exhibit grokking, runs with 90% or 90% of the data do not exhibit grokking”

C) Titles of some appendix plots are cut.





**Summary Of The Paper:**

**Update after rebuttal** The authors have addressed my comments, and, in my view, the issues raised by the other reviewers to a satisfying degree. Given all this information I remain strongly convinced that this paper should be presented at the conference.

The paper investigates grokking on a modular addition task. As has been previously reported, training a large enough transformer on this task leads to a sudden jump from very low test-set accuracy to very high test accuracy, long after training error has converged to essentially zero. The phenomenon can be produced robustly, but has so far been not very well understood. The paper first identifies the algorithm implemented by the network to strongly generalize beyond the training set. Based on this insight, two specific metrics (progress measures) are developed that can be used to diagnose the onset and transition from a memorizing solution to the grokking solution. Using these measures the paper sheds some light on the grokking phenomenon by identifying three phases (memorization, circuit formation where the generalizing solution smoothly starts to form, and cleanup where the grokking solution takes over and weight decay pushes the memorizing weights towards zero).

**Summary Of The Review:**

The paper studies a timely and potentially very interesting phenomenon: the phenomenon that sometimes (transformer) networks can learn (simple) algorithms that lead to strong generalization beyond the training data, despite initially completely overfitting the training data (memorizing). The paper adds to this by performing an excellent case study of analyzing one such solution in great detail; ultimately identifying the algorithm learned by the network. To me personally, this analysis itself is enough to warrant publication - it is another piece of work showing that reverse-engineering the algorithms implemented by neural networks is actually possible. Based on this analysis two diagnostic metrics for continuously characterizing grokking during training are developed, which provide very insightful results (onset of circuit formation after memorizing, switching over to grokking solution only after circuit is well developed but then with fairly rapid transition). These results, and the methodology introduced (which is admittedly quite labor intensive) open up the venue for studying the precise conditions under which the grokking solution is formed more generally; which might lead to improved training methods that robustly produce strongly generalizing solutions. I do not have any major concerns, and left some small comments and suggestions for improvements. Currently I am in favor of accepting the paper at the conference - I think it is of great interest to large parts of the community.

---

> ### Author Response · Authors · 2022-11-17
> **Response to reviewer u853**
>
> Thanks for your incredibly detailed review and feedback! We’ve fixed the typos and graphical errors and also added your suggestion A) on “capabilities” to the first line of the intro.
>
> > The current discussion could be a bit expanded and perhaps also be a bit more critical - the intro sounds a somewhat like the current paper is a big leap towards general/task-independent progress measures that allow predicting the onset of arbitrary capabilities (yet the current paper highlights the challenges which make such general measures unlikely to exist).
>
> This is a good point. We agree that our work does highlight several challenges to finding general progress measures, and also that our work does not show that mechanistic interpretability can succeed at finding progress measures to predict arbitrary tasks. Instead, we see the work as describing a possible approach, demonstrating the approach works on a task, and illuminating some of the future work required to scale to more general progress measures. We’ve edited a few sentences in the intro and discussion to more accurately reflect this.
>
> > Does training (with different random seeds) always identify the same key frequencies? If yes, is there a hypothesis why that is the case; if no, please mention it and comment on this being a limitation for the design of more general progress measures.
>
> The frequencies learned are random and vary across random seeds—there isn’t a fundamental reason why the model should bias toward particular frequencies, as long as it has sufficiently diverse pool of them (for the constructive interference reason discussed above).
>
> Specifically, in order to compute the `restricted` and `excluded` losses for other models, we perform a DFT of the logits (or almost any other part of the network) and look at the non-trivial components to recover the key frequencies, in the same way we figured out the key frequencies of the mainline network. We admit that this is a limitation to our approach as stated. That being said, we can derive a random seed-independent progress measure by calculating the excluded loss for each of the 56 single different frequencies, and then taking the max over this set.
>
> We’ve added some discussion of this to the paper, and added results to the appendix that show the frequencies of (and demonstrate the reproducibility of our results on) networks learned from other random seeds.
>
>
> > Suggestion (but given the time-frame I do not expect to see these results): identify the “winning lottery ticket” before and after grokking (i.e. remove any unnecessary weights from the network). (I) What is the size of these tickets (and how does it change during circuit formation and cleanup) - related to Fig. 6 bottom right? (II) If the winning ticket after grokking is re-initialized with its original initialization values (and all other parameters are kept at zero during training), can the network recover the grokking solution (without going through a memorization phase; perhaps even faster than in the original training run; in other words: is it enough to identify the correct sub-circuit structure for the grokking solution or do we need the memorization-part of the network during training such that the grokking solution can develop)?
>
> This is a very interesting experiment, so we are currently exploring this angle. We currently find a lottery ticket-based explanation for how the generalizing circuit forms compelling, but we’re unsure as to whether the memorization part of the network is necessary for grokking. While we don’t expect the results before the end of the review period, we will update the paper with any interesting results in this direction. Thanks!

---

> > ### Author Response · Authors · 2022-11-17
> > **Response to reviewer u853, continued**
> >
> >
> > > No need to add much, but it would be great to see a slightly expanded critical discussion (particularly the last two points). What does the current paper contribute to the development of (general) progress measures, and how hard/easy are the current obstacles?
> >
> > As previously stated, the primary contributions of this paper are 1) highlighting an approach (progress measures via mechanistic interpretability) that might be promising for predicting or explaining emergent behavior, 2) demonstrating the approach on an algorithmic task, and 3) using the fruits of this approach to understand grokking on this task.
> >
> >
> > We think there are several clear obstacles toward using mechanistic interpretability to derive progress measures for models in practice, which we mention in the discussion. The first is a need for more scalable mechanistic interpretability techniques that work on larger networks, as we mention in the conclusion. The second is an in-depth understanding of the thresholds by which continuous progress measures lead to discontinuous downstream performance.
> >
> > Generally speaking, we think there are three approaches for extending this work to using progress measures to understand emergent phenomena:
> > 1. If mechanistic interpretability can be scaled to large models to the level where we can understand the mechanisms behind significant portions of their behavior, we could perform the same style of analysis as was done in this work, and use the parts of the mechanism as a progress measure.
> > 2. If future mechanistic interpretability can only recover parts of the mechanism of larger models and can only generate comprehensive understanding of the mechanisms of smaller models, we might still be able to use our understanding from smaller models to guide the development measures that track parts of the behavior of the larger model. For example, Olsson et al 2022 do this to understand the emergence of in-context learning in medium-sized language transformers.
> > 3. Even if mechanistic interpretability fails to recover understandable mechanisms at all on large models, we might still be able to derive progress measures that don’t require human understanding. For example, if we end up with automated mechanistic interpretability (that nonetheless still fails to recover human-understandable mechanisms), we might be able to use the outputs of those opaque processes. Another approach is task-independent progress measures: for example, we’ve attempted some preliminary experiments attempting to derive progress measures by looking at the principle components of variation of the network weights and logits over time*; if approaches like these worked, we might be able to derive task-independent progress measures. The lottery ticket style of approach you suggest might also allow us to derive task-independent progress measures.
> >
> > We’ve edited the discussion slightly as suggested and have added a section in the appendix expanding on the points we mention above.
> >
> >
> > =======
> >
> > *The two preliminary experiments:
> > 1) We took a checkpoint from every 100 epochs and flattened the model weights, which we then concatenated into a [400, # params] matrix consisting of the model weights over the course of training. We then performed a PCA on this matrix and found that the top 4 principal components explain the majority of the variation of the weights. We found that the first and second components increased drastically immediately after training began, while the third and fourth principal components were relatively stable until they both increased significantly around the grokking phase. We’re hoping to better understand these components in order to understand how the model performs the interpolation.
> >
> > 2) We took a checkpoint from every 100 epochs, used each checkpoint to compute the logits (a rank-3 tensor of dimensions (113, 113, 113)), flattened the logits, and then concatenated them into a [400, 113 x 113 x 113] matrix. We then plotted the first two principal components of the matrix of flattened logits over the course of training. (Results here: https://imgur.com/a/YIZu7Cr). We found that the model seems to go linearly in one direction until epoch ~1.4k, swerves and then goes linearly in a different direction until epoch ~10k, at which point it swerves again until ~14k where it holds relatively steady. Unfortunately, the first two principal components do not capture a significant fraction of the variance (for example, while there seems to be one “memorizing” direction, it turns out there are [unsurprisingly] 5 generalizing directions, all of which are significant), which poses a challenge for this approach.

---

> > > ### Comment · Reviewer_u853 · 2022-11-28
> > > **Thanks for the detailed response!**
> > >
> > > Thank you for the detailed response, clarifications, and additional preliminary results. Reading the other reviews and responses I remain clearly in favor of accepting the paper. Just to clarify: what I was getting at in terms of general progress measures is that from an advanced AI safety viewpoint it would be nice to develop progress measures that could inform us about an immediate onset of rapid capability increases. What the current work shows is that such progress measures might be highly task specific (which is fine, it could just mean that we need a large battery of measures) and cannot (easily) be developed in advance but only in hindsight by using a network that has already acquired the capabilities. (no need to add/modify anything, just a clarification).

---

### Official Review · Reviewer_g9KN · 2022-10-25

**Confidence:** 4
**Correctness:** 4
**Technical Novelty And Significance:** 4
**Empirical Novelty And Significance:** 4
**Recommendation:** 8

**Clarity, Quality, Novelty And Reproducibility:**

**Clarity**

The paper is generally well-written and easy to follow. A couple of questions:
* How does the constructive interference in the last step of the algorithm work? I need help understanding how to recover the correct solution given the details in the paper.
* Does the model always learn the same key frequencies when initialized with different seeds? Otherwise, how can one compute the `restricted` and `excluded` losses without knowing the values of the key frequencies?

**Quality**

The analysis is both rigorous and insightful.

**Novelty**

To the best of my knowledge, the paper is the first to reverse-engineer the grokking phenomenon for transformers on modular addition fully.

**Reproducibility**

The paper provides sufficient details of the experimental setup (including code and interactive plots) to reproduce the results.

**Strength And Weaknesses:**

**Strengths**

The paper presents a landmark study on the mechanistic interpretability of transformers. It prevents strong empirical evidence (approximation and ablation) that transformers learn to perform modular addition via Fourier multiplication. Thus, even though the paper does not formally prove the model (which is not the goal of this work), it sets an example in conducting a rigorous analysis of a model's behavior. Discovering the Fourier multiplication algorithm by looking at the network's weights requires a significant cognitive leap and careful analysis. Moreover, an understated aspect of the paper is that it simplifies the grokking setup (i.e., one layer, only modular addition with a small prime, etc.) to make it amenable to analysis while maintaining the relevant grokking phenomenon. Finally, the paper opens many follow-up research directions (see questions below).


**Weaknesses**

The paper appears to be somewhat hastily written, as there are a couple of (minor) mistakes:
* Introduction: `that detect earlier repeated subsequences` is redundant.
* Related Work: `toward the generalizing algorithm*.* Liu et al. (2022)`.
* Section 4.2: `should ~contain~ be rank 10`.
* Section 4.2: `yet we can *approximate* the logits`.
* Section 4.3: What does `we find that the attention pattern of the = to a for each of the four heads` mean?
* Section 4.4: `depend only *on* these 10 directions`.
* Section 5.1: `it makes ~since~*sense* to check`.


**Questions**

Note that these are mostly follow-up research questions that do not have to be answered in the context of this work.

* Does grokking also occur with sinusoidal positional encodings, which should be compatible with the Fourier multiplication algorithm?
* How does the model effectively interpolate between the memorizing and generalizing solutions? Are these implemented as two separate subcircuits?
* Do other regularization methods lead to different generalization algorithms? Fourier multiplication and weight decay are closely linked due to the sparsity of the required algorithm.

**Summary Of The Paper:**

The paper investigates grokking, i.e., the abrupt phase change in the performance of transformers when trained on simple algorithmic tasks, through the lens of mechanistic interpretability. It uses the insights gained to propose new progress measures that vary smoothly throughout training and thus provide new insights into the different training phases. Concretely, the paper provides empirical evidence to show that transformers learn to perform modular addition of two numbers via rotation around a circle, i.e., Fourier multiplication. The paper shows that various components of the learned network can be approximated with the different stages of the Fourier multiplication algorithm. Furthermore, the paper conducts ablation studies to confirm the faithfulness of the proposed approximations. Finally, the paper combines these insights to show that the network first memorizes the training data, then smoothly interpolates between memorization and the (sparse) Fourier algorithm (driven by weight decay), and finally removes the memorization component.

**Summary Of The Review:**

Given the importance of the results, the rigor of the analysis, and the potential impact of the work (see strengths above), I recommend acceptance.

---

> ### Author Response · Authors · 2022-11-17
> **Response to reviewer g9KN**
>
> Thanks for the detailed and thoughtful response!
>
> We’ve corrected the minor mistakes and typos you’ve pointed out in the weaknesses section.
>
> > Section 4.3: What does we find that the attention pattern of the = to a for each of the four heads mean?
>
> The transformer was fed a sequence of three tokens consisting of a, b, and a special “=” token, above which we read the output. In section 4.3, we look at the scores of the 4 attention heads from the third sequence position (the "=" token, using this as the query) to the first sequence position (the "a" token, using this as the key). We’ve edited section 4.3 to better clarify what we did in this subsection.
>
> > Does grokking also occur with sinusoidal positional encodings, which should be compatible with the Fourier multiplication algorithm?
>
> Grokking does occur on this task with both sinusoidal and absolute positional encodings. However, the specific form of the position embeddings does not matter in this case, as 1) there are exactly three positions in every sequence,  2) addition is commutative and the special “=” symbol lets the model distinguish the output sequence position from the sequence positions above a and b, and 3) the fourier algorithm depends on embeddings that are sinusoidal over the vocab dimension, not over the residual stream dimension. That is, the model doesn’t need to use any position embeddings at all to successfully complete the task.
>
> > How does the model effectively interpolate between the memorizing and generalizing solutions? Are these implemented as two separate subcircuits?
>
>
> We think this is a great question, and unfortunately one that we don’t have a good answer to. We hope to answer this question in future work.
>
> (We’re also exploring the lottery ticket experiment suggested by reviewer u853, though we don’t expect to see results before the end of the author discussion period.)
>
> > Do other regularization methods lead to different generalization algorithms? Fourier multiplication and weight decay are closely linked due to the sparsity of the required algorithm.
>
> We’re currently running these experiments and will update the appendix with the answer (and post a reply here) when we get the results.
>
> Note that prior work (specifically, Powers et al 2022) has found grokking on algorithmic tasks for a variety of regularizations, including different variants of weight decay, dropout, and weight noise. We do expect to see grokking and suspect that the generalization algorithm should be similar.
>
> > How does the constructive interference in the last step of the algorithm work? I need help understanding how to recover the correct solution given the details in the paper.
>
> Consider the function $f_{14}(x) = cos\left (\frac{2 \pi \cdot 14}{113} x \right)$. This function has period 113, and is maximized at $x = 0 \mod 113$. However, it turns out that there are other values of x which causes this function to be close to 1: $f_{14}(8) = f_{14}(105) = 0.998$, $f_{14}(16) = f_{14}(105) = 0.994$, etc.
>
> Now consider $f_{35}(x) = cos \left (\frac{2 \pi \cdot 35}{113} x \right)$. While this function also has period 113 and is maximized at $x = 0 \mod 113$, it turns out that $f_{35}(8) = f_{35}(105) = -0.990$. This means that by adding together $f_{14}$ and $f_{35}$, we end up with a function that is not close to 1 at $x=8 \mod 113$. Similarly, while $f_{35}(16) = 0.961$, $f_{52}(16)  = -0.56$, and so adding a third frequency reduces the peak at $x = 16 \mod 113$.
>
> Here’s a plot showing this phenomena: https://imgur.com/a/LRVgbPI
>
> We’ve added some clarification of this point to the appendix.
>
> > Does the model always learn the same key frequencies when initialized with different seeds? Otherwise, how can one compute the restricted and excluded losses without knowing the values of the key frequencies?
>
> The frequencies learned are random, and there isn’t a fundamental reason why the model should bias toward particular frequencies, as long as it has a diverse enough set of them (for the constructive interference reason discussed above).
>
> In order to compute the `restricted` and `excluded` losses for other models, we perform a DFT of the logits (or almost any other part of the network) and look at the non-trivial components to recover the key frequencies, in the same way we figured out the key frequencies of the mainline network. We admit that this is a limitation to our approach as stated. That being said, we can derive a random seed-independent progress measure by calculating the excluded loss for each of the 56 single different frequencies, then taking the max over this set. We’ve added some discussion of this to the appendix.

---

### Author Response · Authors · 2022-11-17
**Summary of updates for rebuttal revision**

We’d like to thank the three reviewers for their detailed reviews. We’re happy that the reviewers found our paper well-written and were excited by both our mechanistic interpretation of the transformer and use of progress measures to study grokking. We also appreciate the suggestions on clarity and on additional work to strengthen our results. Our paper has been updated based on the feedback.

Here are the changes we’ve made to the paper:
1. We’ve fixed the typos and clarity issues mentioned in the reviews.
2. We’ve added some clarification about why the network performs constructive interference to the appendix, based on the question asked by reviewer g9KN.
3. We’ve added additional results from more training runs to the appendix. In particular, we’ve included replications of much of Section 4 for other training runs with the same architecture and different random seeds.
4. We’ve clarified that the key frequencies are arbitrary and vary across random seeds. We’ve also included the key frequencies for other training runs in the appendix.
5. We’ve introduced a new progress measure: the Gini coefficient of the Fourier components of the embedding and output matrices, which tracks the sparsity of said components. We find that the resulting curves can also be divided into the same three phases.
6. We’ve rescaled our Fourier components. Previously, our Fourier components were normalized such that $\ell_2$ norm of each component was 1 (that is, we divided the components by the normalization factor $\sqrt{113/2}$), and we’ve removed this normalization factor. This change does not affect any of our results, but merely makes the numbers involved smaller.
7. We’ve added a section to the appendix where we discuss some theories for how the network develops the complicated generalizing circuit.
8. We’ve slightly adjusted the introduction and discussion to more accurately reflect the challenges of applying mechanistic interpretability and progress measures to understand emergent phenomena in general. We’ve also added a section to the appendix where we discuss three ways in which this approach can “play out”, depending on the success of future mechanistic interpretability work.
9. We added some new experiments with alternative regularization methods to the appendix.

---

### Decision · Program_Chairs · 2023-01-20

**Decision:**

Accept: notable-top-25%

**Justification For Why Not Higher Score:**

Because this is a relatively limited proof-of-concept paper, it would not be appropriate for an oral, I feel.

**Justification For Why Not Lower Score:**

The reviewers were in clear agreement that this paper is interesting, well-done, and tackles an important question. I therefore feel that this is worthy of a spotlight.

**Metareview: Summary, Strengths And Weaknesses:**

This paper is interested in the question of whether apparently sudden, emergent, qualitative changes in behaviour in neural networks can be explained via continuous progress measures in order to understand what is actually being learned by the networks in these moments. The paper focuses on the specific case of "grokking" in transformers trained on modular addition. Starting with empirical evidence to show that transformers learn to use Fourier multiplication for this task, the authors develop progress measures that show there are stages of training that unfold gradually, demonstrating that the apparently discontinuous moment of grokking is a result of gradual amplification and removal of core underlying operations, such as memorization.

The reviewers all agreed that this is an interesting paper that provides a clear proof-of-concept for how we can better understanding qualitative shifts and emergent phenomena in neural networks. There was consensus that the paper should be accepted.

There were a few minor weaknesses noted, most importantly, that though there is some discussion about how to extend this to more general cases, there is no clear demonstration that it could. Thus, it is a relatively specific proof-of-concept study.

**Note From Pc:**

if the above contains the word "oral" or "spotlight" please see: "oral" presentation means -> notable-top-5% and "spotlight" means -> notable-top-25%. As stated in our emails, we are disassociating presentation type from AC recommendations

**Summary Of Ac-Reviewer Meeting:**

N/A